# INTRINSIC LORA: A Generalist Approach for Discovering Knowledge in Generative Models

Xiaodan Du[1]    Nicholas Kolkin[2]    Greg Shakhnarovich[1]    Anand Bhattad[1]

[1]Toyota Technological Institute at Chicago    [2]Adobe

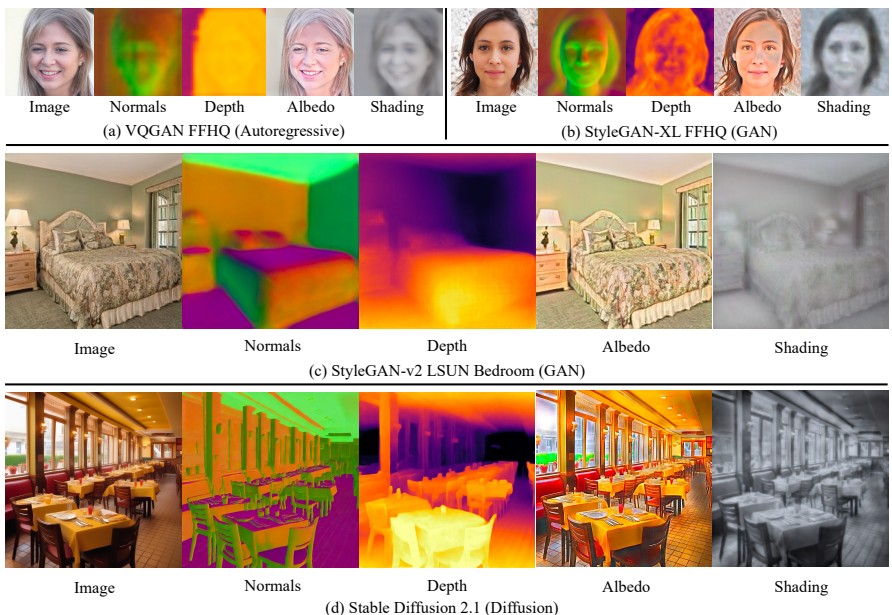

Figure 1. INTRINSIC LORA (I-LORA) is a general approach for extracting visual knowledge from generative models of many types. Our method applies targeted, lightweight fine-tuning to modulate key feature maps, using low-rank adaptation (LoRA) on attention layers in VQGAN (a) and Stable Diffusion (d), and affine layers in StyleGAN (b and c). This process helps us discover fundamental scene intrinsics – normals, depth, albedo, and shading – directly from the models' learned representations, avoiding the need for additional task-specific design of decoding heads or layers.

## Abstract

*Generative models have been shown to be capable of creating images that closely mimic real scenes, suggesting they inherently encode scene representations. We introduce* INTRINSIC LORA *(*I-LORA*), a general approach that uses Low-Rank Adaptation (LoRA) to discover scene intrinsics such as normals, depth, albedo, and shading from a wide array of generative models.* I-LORA *is lightweight, adding minimally to the model's parameters and requiring very small datasets for this knowledge discovery. Our approach, applicable to Diffusion models, GANs, and Autoregressive models alike, generates intrinsics using the same output head as the original images.*

## 1. Introduction

Generative models can produce high-quality images almost indistinguishable from real-world photographs. They seem

Table 1. Summary of scene intrinsics found across different generative models without changing generator head. ✓: Intrinsics can be extracted with high quality. ∼: Intrinsics cannot be extracted with high quality. ✗: Intrinsics cannot be extracted.

| Model | Pretrain Type | Domain | Normal | Depth | Albedo | Shading |
|---|---|---|---|---|---|---|
| VQGAN [14] | Autoregressive | FFHQ | ∼ | ∼ | ✓ | ✓ |
| SG-v2 [27] | GAN | FFHQ | ✓ | ∼ | ✓ | ✓ |
| SG-v2 [54] | GAN | LSUN Bed | ✓ | ✓ | ✓ | ✓ |
| SG-XL [44] | GAN | FFHQ | ✓ | ∼ | ✓ | ✓ |
| SG-XL [44] | GAN | ImageNet | ✗ | ✗ | ✗ | ✗ |
| SD-UNet (single-step) [42] | Diffusion | Open | ✓ | ✓ | ✓ | ✓ |
| SD (multi-step) [42] | Diffusion | Open | ✓ | ✓ | ✓ | ✓ |

to demonstrate a profound understanding of the world, capturing nuances of realistic object placement, appearance, and lighting conditions. Yet, it remains an open question how these models encode such detailed knowledge, and whether representations of scene intrinsics exist in these models and can be extracted explicitly.

**Our Contribution.** We conduct our inquiry across a spectrum spanning diffusion, GANs, and autoregressive models – to understand whether they encode fundamental scene in-

trinsics of normals, depth, albedo, and shading [3]. Our method, INTRINSIC LoRA (I-LoRA), a Low-Rank Adaptation (LoRA) technique, efficiently extracts these intrinsics across different model types with minimal computational overhead and data requirements. Detailed results and a summary are presented in Tab. 1 and elaborated further in Sec. 4. Our experiments suggest that the intrinsic knowledge within generative models is not accidental but a byproduct of large-scale learning to mimic image data. In summary, our work broadens the understanding of visual knowledge within generative image models and our contributions are:

- **Wide Applicability:** We validate I-LoRA 's capability to extract scene intrinsics (normals, depth, albedo, and shading) across a broad spectrum of generative models, highlighting its adaptability to diverse architectures.
- **Efficient and Lean Approach** to knowledge extraction: I-LoRA is highly efficient, requiring a little increase in parameters (less than 0.17% for Stable Diffusion) and minimal training data, as few as 250 images.
- **Insights from Learned Priors:** Through control experiments, we illustrate the critical role of learned priors, suggesting the quality of intrinsics extracted is correlated to the visual quality of the generative model.
- **Competitive Quality of Intrinsics:** Our method, supervised with hundreds to thousands of labeled images, generates intrinsics on par with or even better than those produced by the leading supervised techniques requiring millions of labeled images.

## 2. Related Work

**Generative Models:** Generative Adversarial Networks (GANs) [17] have been widely used for generating realistic images. Variants like StyleGAN [25], StyleGAN2 [27] and GigaGAN [23] have pushed the boundaries in terms of image quality and control.

Diffusion models, such as Denoising Score Matching [49] and Noise-Contrastive Estimation [18], have been used for generative tasks and are perhaps the most popular at the moment [20, 28, 42].

Autoregressive models like PixelRNN [47] and Pixel-CNN [46] generate images pixel-by-pixel, offering fine-grained control but at the cost of computational efficiency. More recently, VQ-VAE-2 [41] and VQGAN [14] have combined autoregressive models with vector quantization to achieve high-quality image synthesis.

**Scene Intrinsics Extraction:** Barrow and Tenenbaum [3] highlighted several fundamental scene intrinsics including depth, albedo, shading, and surface normals. A large body of work has focused on extracting some related properties, like depth and normals from images [4, 12, 13, 24, 32, 40] using labeled annotated data. Labeled annotations of albedo and shading are hard to find and as the recent review in [15] shows, methods involving little or no learning have remained

competitive until fairly recently. However, these methods often rely on supervised learning and do not explore the capabilities of generative models in this context.

Many recent studies have used generative models [1, 2, 22, 29, 30, 37, 43, 51, 57, 58] as pre-trained feature extractors or scene prior learners. They use generated images to enhance downstream discriminative models, fine-tune the original generative model for a new task, learn new layers or decoders to produce desired scene intrinsics.

**Knowledge in Generative Models:** Several studies have explored the extent of StyleGAN's knowledge, particularly in the context of 3D information about faces [38, 56]. Yang et al. [52] show GANs encode hierarchical semantic information across different layers. Further research has demonstrated that manipulating offsets in StyleGAN can lead to effective relighting of images [5] and extraction of scene intrinsics [7]. Chen et al.[9] found internal activations of the LDM encode linear representations of both 3D depth data and a salient-object / background distinction. Recently, [19, 34, 45] found correspondence emerges in image diffusion models without any explicit supervision.

**LoRA (Low-Rank Adaptation).** LoRA [21] introduces trainable low-rank decomposed matrices into specific layers of the model architecture. These matrices are the only components updated during task-specific optimization. This results in a significant reduction in the number of trainable parameters, ensuring only slight modifications to the model, and preserving its core functionality and accessibility.

## 3. INTRINSIC LoRA

A generative model $G$ maps noise/conditioning information $z$ to an RGB image $G(z) \in \mathbb{R}^{H \times W \times 3}$. We seek to augment $G$ with a small set of parameters $\theta$ that allow us to produce, using the same architecture as $G$, an image-like map with up to three channels, representing scene intrinsics.

**I-LoRA's Learning Framework.** Our method, I-LoRA, learns to extract intrinsic properties of an image (such as depth) using a small number of labeled examples (image/depth map pairs) as supervision. In cases where we do not have access to the actual intrinsic properties, we use models trained on large datasets to generate estimated intrinsics as pseudo-ground truth, used as training targets for $G_\theta$.

To optimize $\theta$ of $G_\theta$ using a pseudo-ground truth predictor $\Phi$ (e.g., a network trained to predict depth from an image), we minimize the objective:

$$\min_{\theta} \mathbb{E}_z[d(G_\theta(z), \Phi(G(z)))], \qquad (1)$$

where $d$ is the distance metric.

Diffusion models require special treatment since they are effectively image-to-image and not noise-to-image. During inference, diffusion models repeatedly receive a noisy image as input. Thus instead of conditioning noise $z$ we feed an

image $x$ (generated or real) to a diffusion model $G$. In this case, given a real image $x$, our objective function becomes $\min_\theta \mathbb{E}_x[d(G_\theta(x), \Phi(x))]$.

For surface normals $\Phi$ is Omnidatav2-Normal [12, 24]. For depth we use ZoeDepth [4] as the predictor $\Phi$. For Albedo and Shading $\Phi$ is Paradigms [6, 15]. For SG2, SGXL and VQGAN, $d$ in Eq.1 is

$$d(x, y) = 1 - cos(x, y) + \|x - y\|_1 \qquad (2)$$

for normal and MSE for other intrinsics. For latent diffusion based methods, there isn't a clear physical meaning to the relative angle of latent vectors in encoded normal maps, so we use the standard objective of MSE for all intrinsics.

We use LoRA to recover image intrinsics from generative models. LoRA introduces a low-rank weight matrix $W^*$, which has a lower rank than the original weight matrix $W \in \mathbb{R}^{d_1 \times d_2}$. This is achieved by factorizing $W^*$ into two smaller matrices $W_u^* \in \mathbb{R}^{d_1 \times d^*}$ and $W_l^* \in \mathbb{R}^{d^* \times d_2}$, where $d^*$ is chosen such that $d^* \ll \min(d_1, d_2)$.

The output $o$ for an input activation $a$ is then given by:

$$o = Wa + W^*a = Wa + W_u^* W_l^* a. \qquad (3)$$

**Applying I-LoRA.** For **GANs**, I-LoRA modules are integrated with the affine layers that map from w-space to s-space [50]. In the case of **VQGAN, an autoregressive model**, I-LoRA is applied to the convolutional attention layers within the decoder. For **diffusion models**, I-LoRA adaptors are learned atop cross-attention and self-attention layers. The UNet is utilized as a dense predictor, transforming an RGB input into intrinsics in one step. This approach, favoring simplicity and effectiveness, delivers superior quantitative results. Depending on the intrinsics of interest, the textual input varies among "surface normal", "depth", "albedo", or "shading".

## 4. Experiments

In this section, we outline I-LoRA's contributions, demonstrating its general applicability across generative models (Sec. 4.1). Control experiments provide evidence of I-LoRA's effectiveness (Sec. 4.2). Note: our analysis in Sec. 4.2 primarily utilizes a single-step I-LoRA model for intrinsic image extraction. In Sec. 5, we discuss the challenge of naively applying I-LoRA to a multi-step Stable Diffusion model. We propose a simple modification to the architecture by adding an extra layer (that is not learned) for improved intrinsic image extraction. We refer to this model as **Augmented I-LoRA** (I-LoRA$_{\text{AUG}}$).

### 4.1. I-LoRA **is General and Universally Applicable**

We evaluate I-LoRA across diverse generative models, including StyleGAN-v2 [55], StyleGAN-XL [44], and VQ-GAN [14], trained on datasets like FFHQ [27], LSUN Bed-

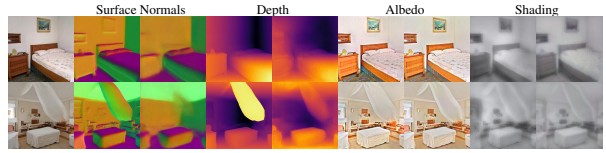
Figure 2. Scene intrinsic properties extracted from StyleGAN-v2 trained on LSUN bedroom images using I-LoRA.

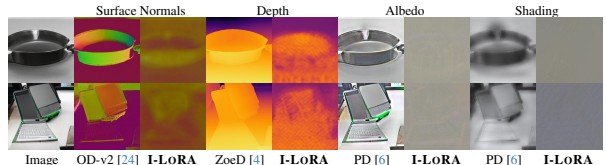
Figure 3. StyleGAN-XL trained on ImageNet. Top: pan, bottom: laptop, with the corresponding scene intrinsics (pseudo ground truth and extracted) alongside. The surface normals and depth maps, while capturing the basic shape and volume, lack precise detail and exhibit artifacts. Albedo and Shading extractions fail. These difficulties are correlated with the overall worse realism and consistency of the generated images.

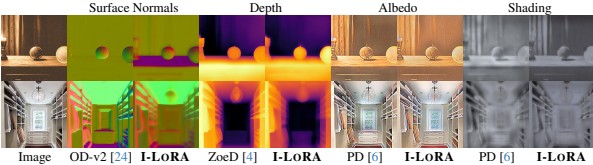
Figure 4. Scene intrinsics from I-LoRA applied to randomly generated images. I-LoRA accurately predicts the table's normal in the first row when compared to [24]. The comparison highlights I-LoRA's ability to closely align with, and sometimes surpass, these supervised SOTA monocular predictors.

rooms [53], and ImageNet [11]. I-LoRA adaptors are tailored to each model and dataset to extract intrinsics: surface normals, depth, albedo, and shading, demonstrating broad applicability and robustness in both qualitative assessments (Fig. 1, 2, 4) and quantitative (Tab. 2 on generated images, Tab. 3 on real images). In all experiments – covering both generated and real images – we use pseudo-ground truth from off-the-shelf models as a supervisory signal. We use I-LoRA with Rank 8 as default for all generative models.

We find I-LoRA can unearth intrinsic knowledge across almost all models tested, the notable exception is StyleGAN-XL trained on ImageNet. Where it yields qualitatively poor results, which we attribute to the model's limited ability to generate realistic images (Fig. 3). This suggests the quality of intrinsic extraction is correlated with the generative model's fidelity (see Sec. 4.2).

In evaluations of generated images, our method is benchmarked against pseudo-ground truths derived from existing models, compensating for the lack of true ground truths.

Diffusion models excel as powerful image generators, thanks to their architecture as image-to-image translators. This feature simplifies their application to real images. Tak-

Table 2. Quantiative analysis of scene intrinsics extraction performance by I-LoRA on generated images. We compare with pseudo GT from Omnidatav2-normal, ZoeDepth and Paradigms.

| Model | Pre-training Type | Domain | LoRA Param. | Surface Normal | | | | Depth | | Albedo | Shading |
|---|---|---|---|---|---|---|---|---|---|---|---|
| | | | | Mean Error°↓ | Median Error°↓ | L1 Error×100↓ | RMS↓ | RMS↓ | δ<1.25×100↑ | RMS↓ | RMS↓ |
| VQGAN | Autoregressive | FFHQ | 0.18% | 19.97 | 20.97 | 16.33 | 0.1819 | 62.33 | | 0.0345 | 0.0106 |
| StyleGAN-v2 | GAN | FFHQ | 0.57% | 16.93 | 19.60 | 13.87 | 0.1530 | 90.74 | | 0.0283 | 0.0110 |
| StyleGAN-XL | GAN | FFHQ | 0.29% | 15.28 | 18.07 | 12.63 | 0.1337 | 93.87 | | 0.0287 | 0.0125 |
| StyleGAN-XL | GAN | LSUN Bedroom | 0.57% | 13.94 | 24.76 | 11.49 | 0.0897 | 66.88 | | 0.0270 | 0.0074 |
| StyleGAN-XL | GAN | ImageNet | 0.29% | 24.09 | 25.52 | 19.44 | 0.2175 | 38.38 | | 0.1065 | 0.0119 |
| I-LoRA_AUG (multi step) | Diffusion | Open | 0.17% | 21.41 | 28.57 | 17.39 | 0.2042 | 41.21 | | 0.0881 | 0.0099 |
| I-LoRA (single step) | Diffusion | Open | 0.17% | 16.63 | 23.64 | 13.69 | 0.1179 | 52.59 | | 0.0487 | 0.0118 |

Table 3. Quantitative analysis of scene intrinsic extraction performance across different models on real images.

| Model | Pre-training Type | LoRA Param | Surface Normal | | | | Depth | |
|---|---|---|---|---|---|---|---|---|
| | | | Mean Error°↓ | Median Error°↓ | L1 Error×100↓ | RMS↓ | RMS↓ | δ<1.25×100↑ |
| Omnidata-v2 [24]/ZoeDepth [4] | Supervised | - | **18.90** | 13.36 | **15.21** | 0.2693 | | **47.56** |
| I-LoRA_AUG (multi step) | Diffusion | 0.17% | 23.74 | 19.08 | 19.31 | 0.2651 | | 43.19 |
| I-LoRA (single step) | Diffusion | 0.17% | 20.31 | **12.54** | 16.53 | **0.2046** | | 44.90 |

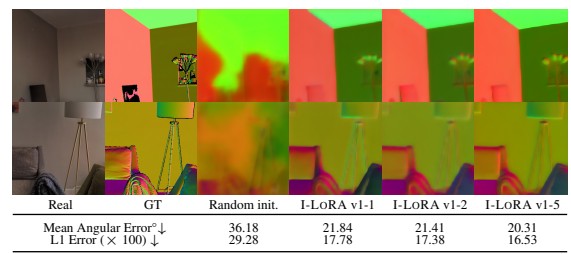

| | Real | GT | Random init. | I-LoRA v1-1 | I-LoRA v1-2 | I-LoRA v1-5 |
|---|---|---|---|---|---|---|
| Mean Angular Error°↓ | | | 36.18 | 21.84 | 21.41 | 20.31 |
| L1 Error (×100)↓ | | | 29.28 | 17.78 | 17.38 | 16.53 |

Figure 5. We find a correlation between generative model quality and scene intrinsic extraction accuracy.

ing advantage of this, we apply I-LoRA to directly extract intrinsic images from Stable Diffusion's UNet in a single step. This method bypasses the iterative reverse denoising process. The model receives a real image as input and outputs the corresponding image intrinsics through I-LoRA. Such direct application allows for evaluation against actual ground truth. This provides a definitive benchmark for assessing I-LoRA's effectiveness (Tab. 3 and Fig. 5). We evaluate on DIODE dataset [48] containing a diverse range of complex indoor and outdoor scenes.

In Tab. 3, we find that I-LoRA not only matches but, in several metrics (surface normals median error, depth RMSE), surpasses the performance of Omnidata and ZoeDepth – the source of its training signal – while using significantly less data, parameters, and training time.

### 4.2. Control Experiments and Correlation with Generative Quality

To assess if our I-LoRA leverages pre-trained generative capabilities or primarily depends on LoRA layers, we performed a control experiment using a randomly initialized SD UNet, following the same training protocol of our I-LoRA model. The poor results from this model, presented in Fig. 5, corroborate that the learned features developed during generative pre-training are crucial for intrinsic extraction, rather than I-LoRA layers alone.

Furthermore, analyzing multiple Stable Diffusion versions (v1-1, v1-2 and v1-5) under the same training protocol reveals that enhancements in image generation quality correlate positively with intrinsic extraction capabilities. This

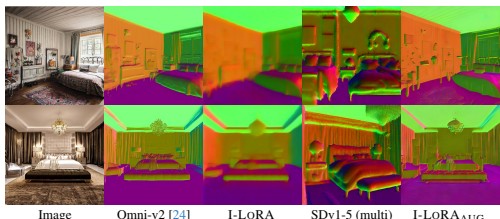

Figure 6. I-LoRA yields satisfactory results, but multiple diffusion steps lead to misalignment in extracted intrinsics, see the SDv1-5 column. The last column, I-LoRA_AUG , demonstrates successfully correcting the misalignment using our image conditioning approach, resulting in well-aligned and detailed intrinsic extractions

assertion is further reinforced by observing a correlation between lower FID scores (9.6 for VQGAN [14], 3.62 for StyleGAN-v2 [26] and 2.19 for StyleGAN-XL [44]) and improved intrinsic predictions in our FFHQ dataset experiments, illustrated in Tab. 2 (first three rows).

## 5. I-LoRA_AUG : Augmented I-LoRA

Can we enhance the quality of the intrinsics by leveraging the multi-step diffusion inference? While multi-step diffusion improves sharpness, we find it introduces two challenges: 1. intrinsics misaligned with input, and 2. shift in the distribution of outputs relative to the ground truth (visually manifesting as a color shift) (see Fig. 6).

To address the first challenge, we augment the noise input to the UNet with the input image's latent encoding, as in InstructPix2Pix [8] (IP2P). The second challenge is a known artifact attributed to Stable Diffusion's difficulty generating images that are not with medium brightness [10, 31]. Following [31], we replace SDv1-5 with SDv2-1 while maintaining our previously described learning protocol. We name this multi-step augmented SDv2-1 model I-LoRA_AUG. I-LoRA_AUG solves the misalignment issue and reduces the color shift significantly (Fig. 6), resulting in the generation of high-quality, sharp scene intrinsics with improved quantitative accuracy. However, quantitatively, the results still fall short of our single-step I-LoRA result. In the future, we hope this problem will be solved by improved sampling techniques and the next generation of generative models.

## 6. Conclusion

In conclusion, we find consistent evidence that generative models implicitly learn scene intrinsics, allowing tiny LoRA adaptors to extract this information with minimal fine-tuning on small labeled data. More powerful generative models produce more accurate intrinsics, strengthening our hypothesis that learning this information is a natural byproduct of learning to generate images well. Finally, we discovered scene intrinsics exist across different generative models, resonating with Barrow & Tenenbaum's hypothesis of fundamental "scene characteristics" emerging in visual processing [3].

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

# Appendices

## A. I-LoRA Pipeline

Fig. 7 illustrates the I-LoRA pipeline applied to Stable Diffusion's UNet in a single-step manner.

## B. Additional Ablation Studies

### B.1. Rank Efficiency

Our single-step I-LoRA model, distinguished by its high quantitative performance, serves as the basis for ablation studies that assess the influence of rank and labeled data quantity on intrinsic extraction efficiency. We verify that the requirements for compute, parameters, and data to learn I-LoRA are minimal.

Fig. 8 shows surface normal predictions across LoRA ranks. The highest accuracy is achieved with Rank 8, balancing accuracy and memory. Notably, a Rank 2 LoRA with only 0.4M additional parameters (a mere 0.04% increase) still yields good performance. Note that across different generative models, Rank 8 adaptors adds only 0.17% to 0.57% additional parameters (Tab. 2).

### B.2. Label Efficiency

The impact of the labeled data size is analyzed in Fig. 9. I-LoRA reaches peak performance using a modest 4000 training examples, with credible predictions visible from as few as 250 samples.

### B.3. Number of Diffusion Steps

To assess the impact of the number of diffusion steps on the performance of the multi-step I-LoRA$_{\text{AUG}}$ model, we conducted an ablation study. The results are presented in Fig. 10. For all our experiments in the main text, we used DPMSolver++ [33]. Interestingly, the quality of results did not vary significantly with an increased number of steps, indicating that 10 steps are sufficient for extracting better surface normals from the Stable Diffusion. Nevertheless, we use 25 steps for all our experiments because it is more stable across different image intrinsics.

### B.4. CFG scales

When working with the multi-step I-LoRA$_{\text{AUG}}$, the quality of the final output is influenced by the choice of classifier-free guidance (CFG) scales during the inference process. In Fig. 11, we present a comparison of the effects of using different CFG scales. Based on our experiments, we found that using CFG=3.0 results in the best overall quality and minimizes color-shift artifacts.

## C. Baselines

### C.1. Superiority of I-LoRA over Fine-tuning and Linear Probing

We compare I-LoRA with two common baselines: linear probing and full model fine-tuning. Following Chen et al.[9] for linear probing and employing standard fine-tuning practices, we train all methods with a small dataset of 250 samples to 16000 samples. Our findings, detailed in Tab. 4 and illustrated in Fig. 12, indicate that I-LoRA significantly outperforms these baselines in low-data regimes, validating its superior efficacy and data efficiency.

### C.2. Other Ablations and Baselines

We extensively study the effect of applying LoRA to different attention layers within Stable Diffusion models. Specifically, we investigate the outcomes of targeting up-blocks, mid-block, down-blocks, cross-attention, and self-attention layers individually. We find (Fig. 13) that isolating LoRA to up or down blocks or the mid-block alone is less effective or diverges, and applying to either cross- or self-attention layers yields decent results, though combining them is best.

Additionally, we evaluated other image editing methods such as Textual Inversion [16] and VISII [36], alongside InstructPix2Pix's response to "Turn it into a surface normal map" instruction [8]. As shown in Fig. 14, these methods perform poorly for intrinsic image extraction, demonstrating the effectiveness of our I-LoRA approach in extracting scene intrinsics.

### C.3. Baseline of Directly Applying SDEdit

In addition to baselines we discussed above, here we show that directly applying SDEdit [35] will also fail to extract reasonable image intrinsics. We take the model from the SDv1-5 column in Fig. 6 of the main paper and apply SDEdit. In Fig. 15, we show directly applying SDEdit results in severe artifacts, regardless of strength.

## D. Hyper-parameters

In Table 5, we show the hyperparameters we use for each model.

## E. Generated Images Used for Quantitative Analysis

In Tab. 2 of the main paper, we report quantitative results on synthetic images. For Autoregressive models and GANs, we first randomly sample 500 noises and use them to generate 500 RGB images. The same 500 noises will then be used to generate intrinsics with our learned LoRAs loaded. For Stable Diffusion experiments (both single-step and multi-step), we use a single dataset with 1000 synthetic images with

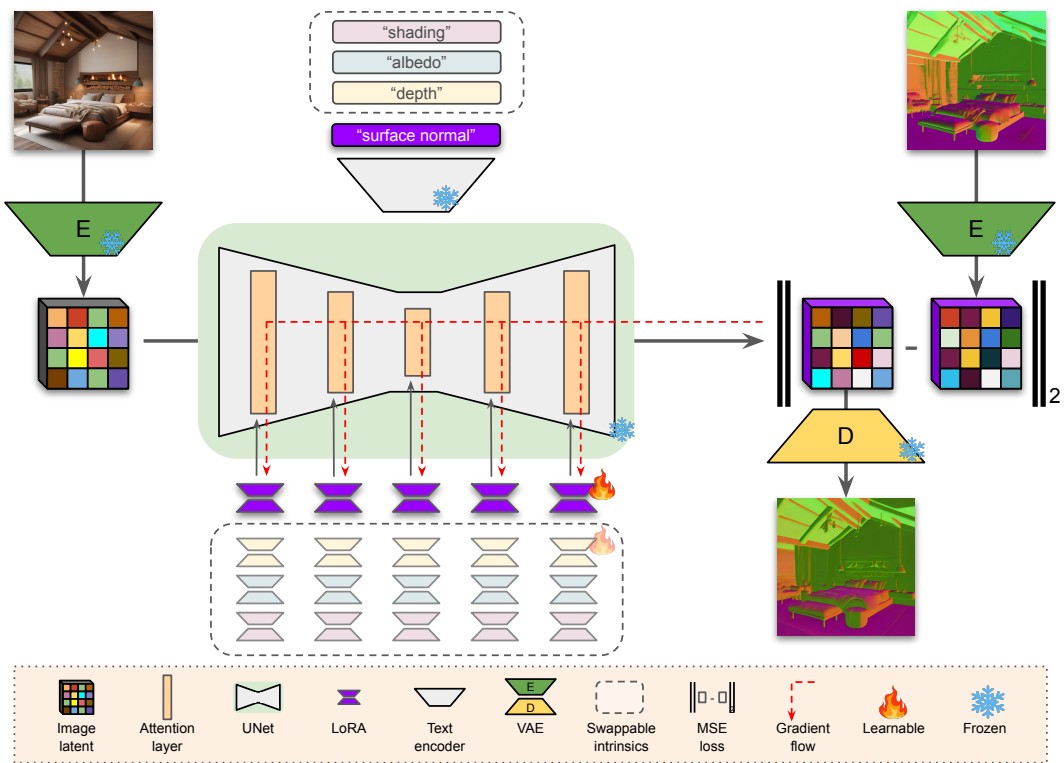

Figure 7. Overview of I-LoRA applied to Stable Diffusion's UNet in a single-step manner. We adopt an efficient fine-tuning approach, specifically low-rank matrices corresponding to key feature maps – attention matrices – to reveal scene intrinsics. Distinct low-rank adaptors (LoRA) are optimized for each intrinsic (*violet* adaptors for surface normals; swappable with other intrinsics). We use a few labeled examples for this fine-tuning and directly extract scene intrinsics using the same decoder that generates images, circumventing the need for specialized decoders or comprehensive model re-training.

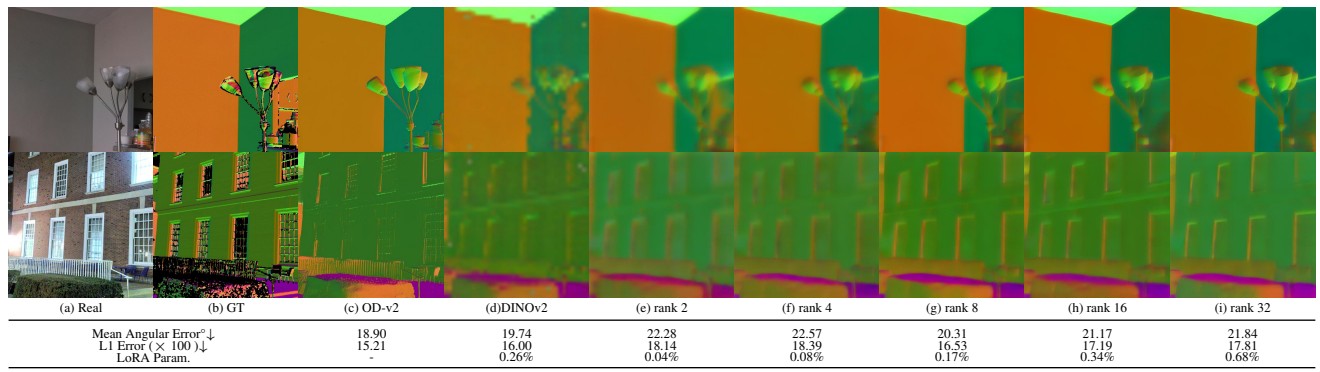

| | (a) Real | (b) GT | (c) OD-v2 | (d)DINOv2 | (e) rank 2 | (f) rank 4 | (g) rank 8 | (h) rank 16 | (i) rank 32 |
|---|---|---|---|---|---|---|---|---|---|
| Mean Angular Error°↓ | | | 18.90 | 19.74 | 22.28 | 22.57 | 20.31 | 21.17 | 21.84 |
| L1 Error (× 100 )↓ | | | 15.21 | 16.00 | 18.14 | 18.39 | 16.53 | 17.19 | 17.81 |
| LoRA Param. | | | - | 0.26% | 0.04% | 0.08% | 0.17% | 0.34% | 0.68% |

Figure 8. Parameter Efficiency of I-LoRA. We evaluate I-LoRA across various rank settings for surface normal extraction. Lower ranks such as 8 offer a balance between efficiency and effectiveness. All model variants are trained using SD's UNet (v1.5) with 4000 samples. Performance metrics, such as Mean Angular Error and L1 Error for normals, and additional parameter counts are detailed below each variant.

Table 4. We find LoRA to consistently outperform all baselines for different number training samples (first row).

| | 250 | | 1000 | | 4000 | | 16000 | |
|---|---|---|---|---|---|---|---|---|
| | Mean Error°↓ | L1 $_{\times 100}$ ↓ | Mean Error°↓ | L1 $_{\times 100}$ ↓ | Mean Error°↓ | L1 $_{\times 100}$ ↓ | Mean Error°↓ | L1 $_{\times 100}$ ↓ |
| Linear Probe | 29.10 | 23.74 | 28.45 | 23.25 | 28.52 | 23.26 | 28.22 | 23.11 |
| Fine-tuning | 34.40 | 27.58 | 25.19 | 20.28 | 28.03 | 22.17 | 27.39 | 22.24 |
| LoRA (Ours) | **27.73** | **22.46** | **22.22** | **18.05** | **20.31** | **16.53** | **21.26** | **17.33** |

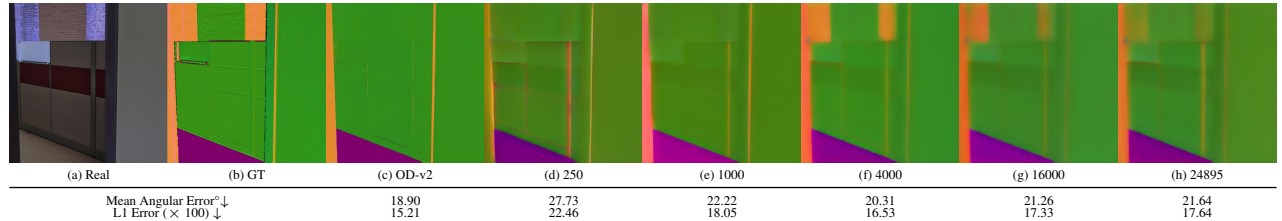

| | | | (a) Real | (b) GT | (c) OD-v2 | (d) 250 | (e) 1000 | (f) 4000 | (g) 16000 | (h) 24895 |
|---|---|---|---|---|---|---|---|---|---|---|

| | (c) OD-v2 | (d) 250 | (e) 1000 | (f) 4000 | (g) 16000 | (h) 24895 |
|---|---|---|---|---|---|---|
| Mean Angular Error° ↓ | 18.90 | 27.73 | 22.22 | 20.31 | 21.26 | 21.64 |
| L1 Error (× 100) ↓ | 15.21 | 22.46 | 18.05 | 16.53 | 17.33 | 17.64 |

Figure 9. Data efficiency of I-LoRA. We report results from varying training samples. Even with 250 samples, I-LoRA captures surface normals. We observe the best performance with 4k samples. Models (d)-(h) all use the same SD UNet(v1-5) and rank 8 LoRA. Note: SOTA supervised model (c), was trained using 12M+ labeled training samples.

| | Omni-v2 [24] | Steps=2 | Steps=5 | Steps=10 | Steps=15 | Steps=20 | Steps=25 | Steps=50 |
|---|---|---|---|---|---|---|---|---|
| Mean Angular Error° ↓ | 25.83 | 23.79 | 23.48 | 23.86 | 23.79 | 23.74 | 23.67 | |
| L1 Error (× 100) ↓ | 21.08 | 19.39 | 19.10 | 19.40 | 19.35 | 19.31 | 19.25 | |

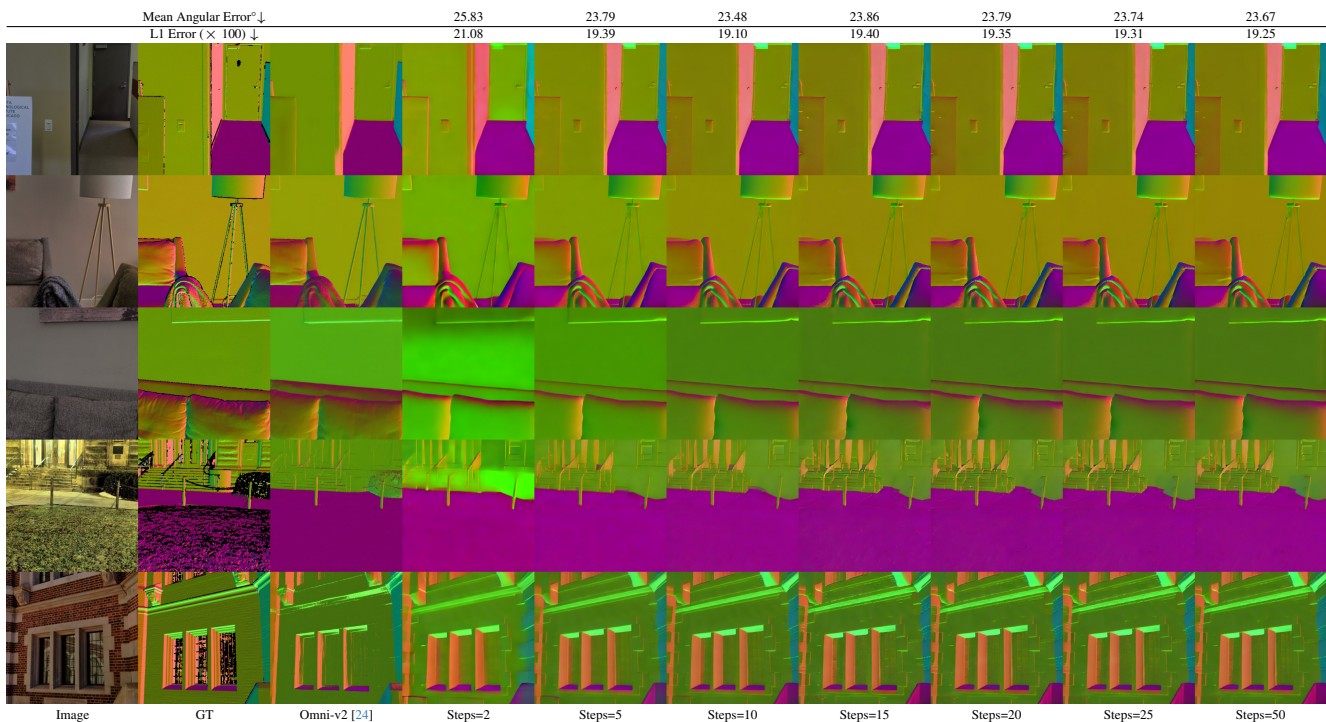

| Image | GT | Omni-v2 [24] | Steps=2 | Steps=5 | Steps=10 | Steps=15 | Steps=20 | Steps=25 | Steps=50 |
|---|---|---|---|---|---|---|---|---|---|

Figure 10. Ablation study to determine the effect of varying numbers of diffusion steps while keeping CFG fixed at 3.0. Our findings show that there are very small differences, both in terms of quantity and quality, after 10 steps. For our main paper, we report results for 25 steps as it is more stable across different intrinsics.

various prompts. The pseudo GT are obtained by applying SOTA off-the-shelf models on the RGB images.

## F. Additional Qualitative Results

In Fig. 16, we present comparisons for I-LoRA$_{AUG}$ and I-LoRA1-5$_{AUG}$. Fig. 17 and Fig. 18 shows extra results for models trained on FFHQ dataset. More examples of scene intrinsics extracted from StyleGAN-v2 trained on LSUN bedroom can be found in Fig. 19. In Fig. 20, we show results for Stable Diffusion I-LoRA (single-step) on generated images. Shown in Fig. 21 are extra results for StyleGAN-XL trained on ImageNet.

## G. Results on $1024^2$ synthetic images

Our multi-step I-LoRA$_{AUG}$ models, although trained exclusively on $512^2$ images from the DIODE dataset, demonstrate their robustness by successfully extracting intrinsic images from $1024^2$ high-resolution synthetic images generated by Stable Diffusion XL [39], as shown across Figures 22 to 31

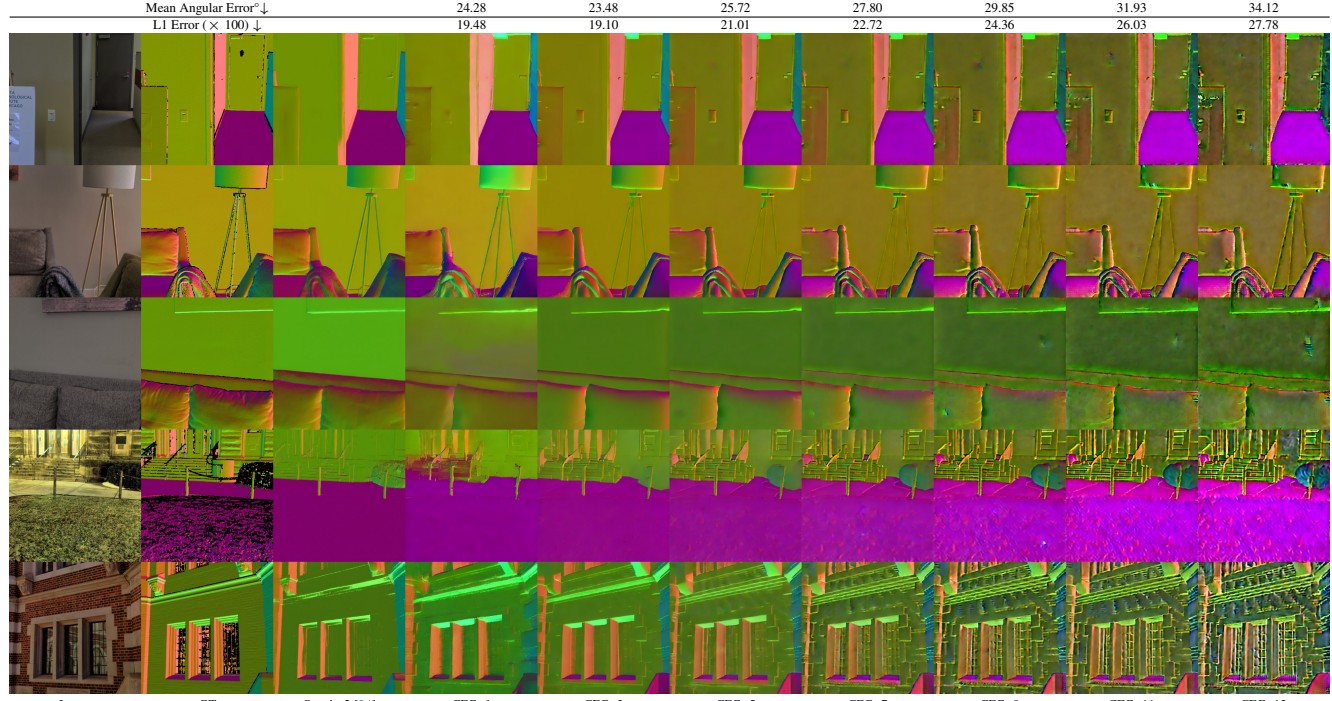

| Mean Angular Error°↓ | | 24.28 | 23.48 | 25.72 | 27.80 | 29.85 | 31.93 | 34.12 |
| L1 Error (× 100) ↓ | | 19.48 | 19.10 | 21.01 | 22.72 | 24.36 | 26.03 | 27.78 |
| Image | GT | Omni-v2 [24] | CFG=1 | CFG=3 | CFG=5 | CFG=7 | CFG=9 | CFG=11 | CFG=13 |

Figure 11. Ablation study analyzing the impact of different classifier-free guidance (CFG) on I-LoRA$_{\text{AUG}}$ surface normal prediction. For efficiency, we experimented with a step of 10. We observed that CFG=1 sometimes led to incorrect semantic predictions, particularly in the case of stairs in row 4. On the other hand, using large CFGs (5 and beyond) results in more severe color shift problems.

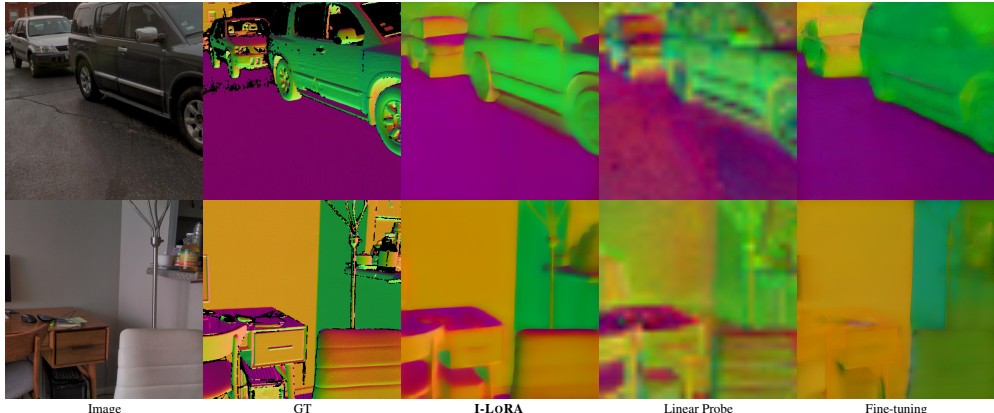

| Image | GT | **I-LoRA** | Linear Probe | Fine-tuning |

Figure 12. Comparison with baselines. All models are trained with 250 samples. Note LoRA effectively extracts better normals compared to other baselines.

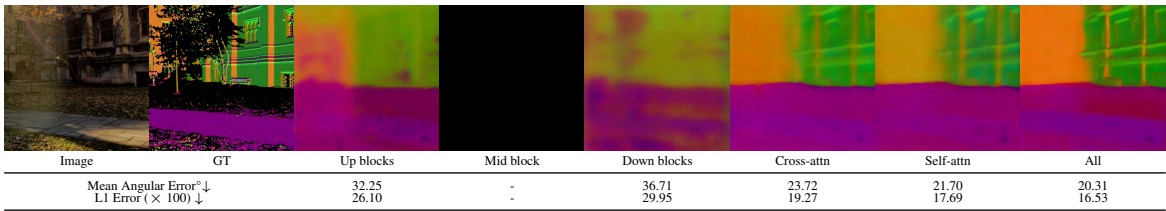

| | Image | GT | Up blocks | Mid block | Down blocks | Cross-attn | Self-attn | All |
| --- | --- | --- | --- | --- | --- | --- | --- | --- |
| Mean Angular Error°↓ | | | 32.25 | - | 36.71 | 23.72 | 21.70 | 20.31 |
| L1 Error (× 100) ↓ | | | 26.10 | - | 29.95 | 19.27 | 17.69 | 16.53 |

Figure 13. Ablation study on the effect of applying LoRA on different types of attention layers. We started all models with SD v1-5, 4000 training samples and LoRA rank=8.

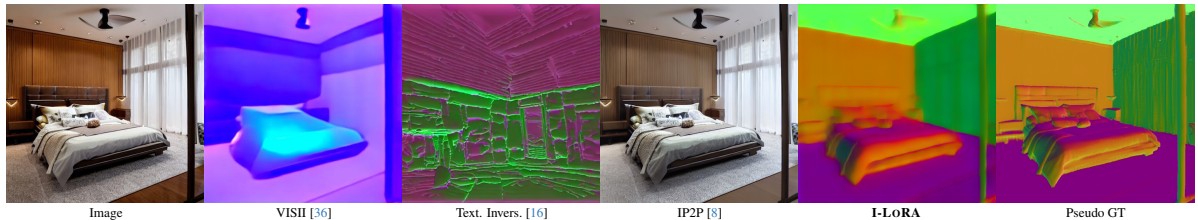

| Image | VISII [36] | Text. Invers. [16] | IP2P [8] | **I-LoRA** | Pseudo GT |

Figure 14. Comparison of image editing techniques for surface normal mapping. VISII and Textual Inversion yield unsatisfactory results, while InstructPix2Pix fails to interpret the task, resulting in near-original output.

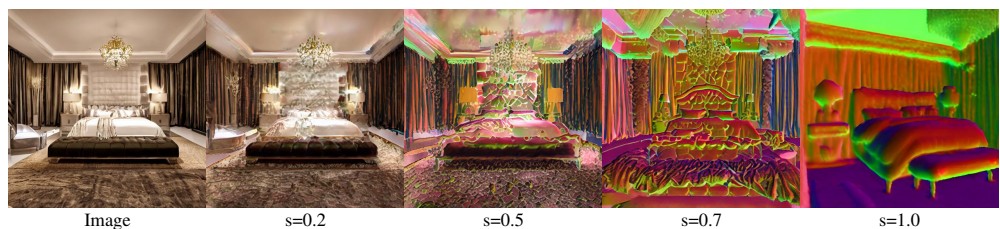

| Image | s=0.2 | s=0.5 | s=0.7 | s=1.0 |

Figure 15. We observe applying SDEdit method on the SDv1-5 model alone, without incorporating the additional input image latent encoding, fails to produce satisfactorily aligned and high-quality scene intrinsics. The reason for this might be the considerable domain shift that exists between RGB images and surface normal maps, which results in severe artifacts when using SDEdit. The variable "s" represents the strength of SDEdit.

| Model | Dataset | Resolution | Rank | LR | BS | LoRA Params | Generator Params | Convergence Steps |
|---|---|---|---|---|---|---|---|---|
| VQGAN | FFHQ | 256 | 8 | 1e-03 | 1 | 0.13M | 873.9M | $\sim$ 4000 |
| StyleGAN-v2 | FFHQ | 256 | 8 | 1e-03 | 1 | 0.14M | 24.8M | $\sim$ 4000 |
| StyleGAN-v2 | LSUN Bedroom | 256 | 8 | 1e-03 | 1 | 0.14M | 24.8M | $\sim$ 4000 |
| StyleGAN-XL | FFHQ | 256 | 8 | 1e-03 | 1 | 0.19M | 67.9M | $\sim$ 4000 |
| StyleGAN-XL | ImageNet | 256 | 8 | 1e-03 | 1 | 0.19M | 67.9M | $\sim$ 4000 |
| I-LoRA$_{AUG}$ (multi step) | Open | 512 | 8 | 1e-04 | 4 | 1.59M | 943.2M | $\sim$ 30000 |
| I-LoRA (single step) | Open | 512 | 8 | 1e-04 | 4 | 1.59M | 943.2M | $\sim$ 15000 |

Table 5. Hyper-parameters for each model. LR refers to the learning rate and BS refers to the batch size. Please note that the number of steps required to reach convergence reported above is for normal/depth. However, it is worth noting that albedo and shading tend to require significantly fewer steps to converge (usually half of normal/depth). Additionally, I-LoRA$_{AUG}$ (multi-step) and I-LoRA (single-step) are trained on real-world DIODE dataset, while the other models are trained on synthetic images within a specific domain. (Num. of params of VQGAN counts transformer + first stage models; Num. of params of I-LoRA$_{AUG}$ and I-LoRA counts VAE+UNet)

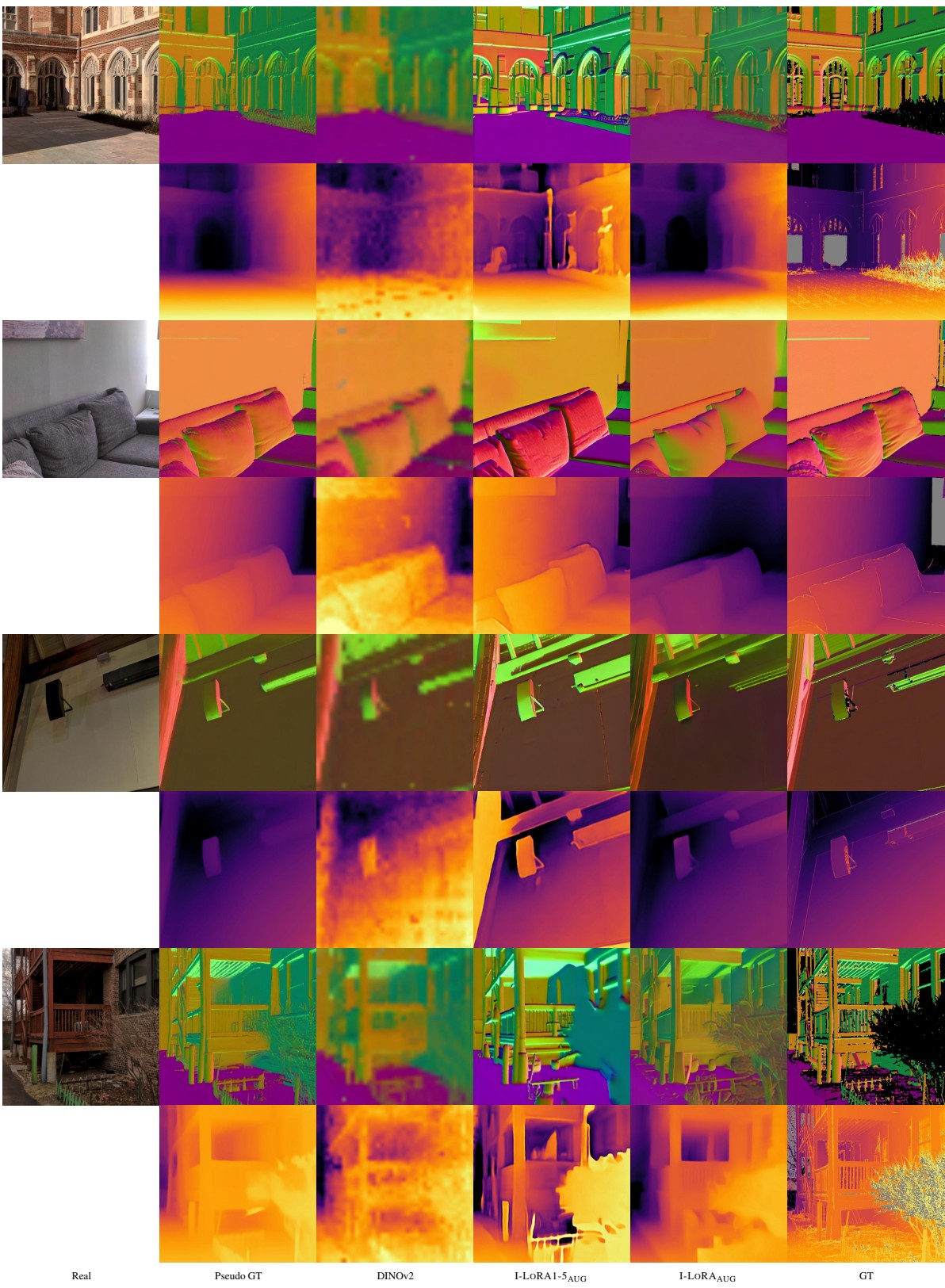

| Real | Pseudo GT | DINOv2 | I-LoRA1-5$_{\text{AUG}}$ | I-LoRA$_{\text{AUG}}$ | GT |

Figure 16. Additional results after applying improved diffusion techniques with I-LoRA$_{\text{AUG}}$. I-LoRA$_{\text{AUG}}$ was found to significantly reduce color shift artifacts observed in I-LoRA1-5$_{\text{AUG}}$ during the extraction of detailed scene intrinsic results.

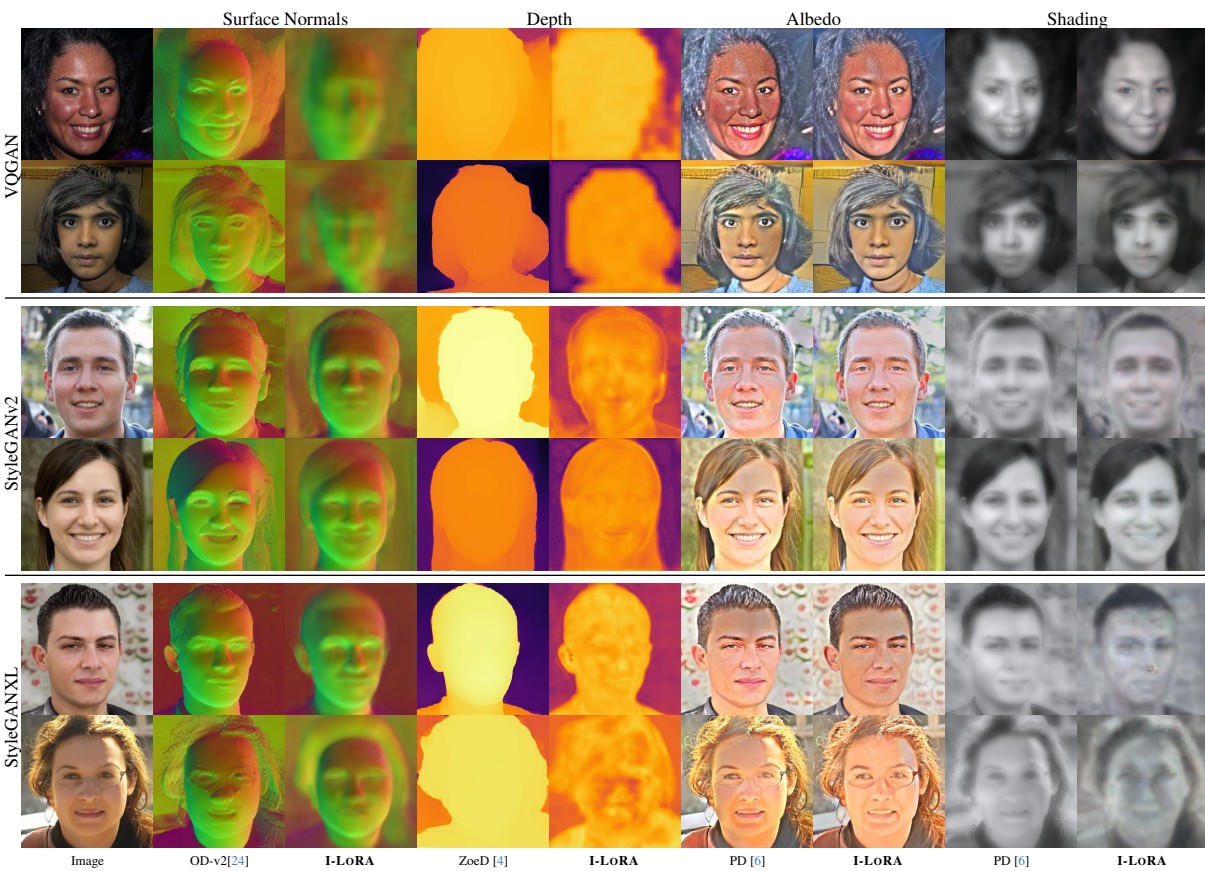

Figure 17. Scene intrinsics from different generators – VQGAN, StyleGAN-v2, and StyleGAN-XL – trained on FFHQ dataset: The "image" column shows the synthetic images produced by each model. Subsequent columns show four scene intrinsics extracted by a SOTA non-generative model and I-LoRA(ours).

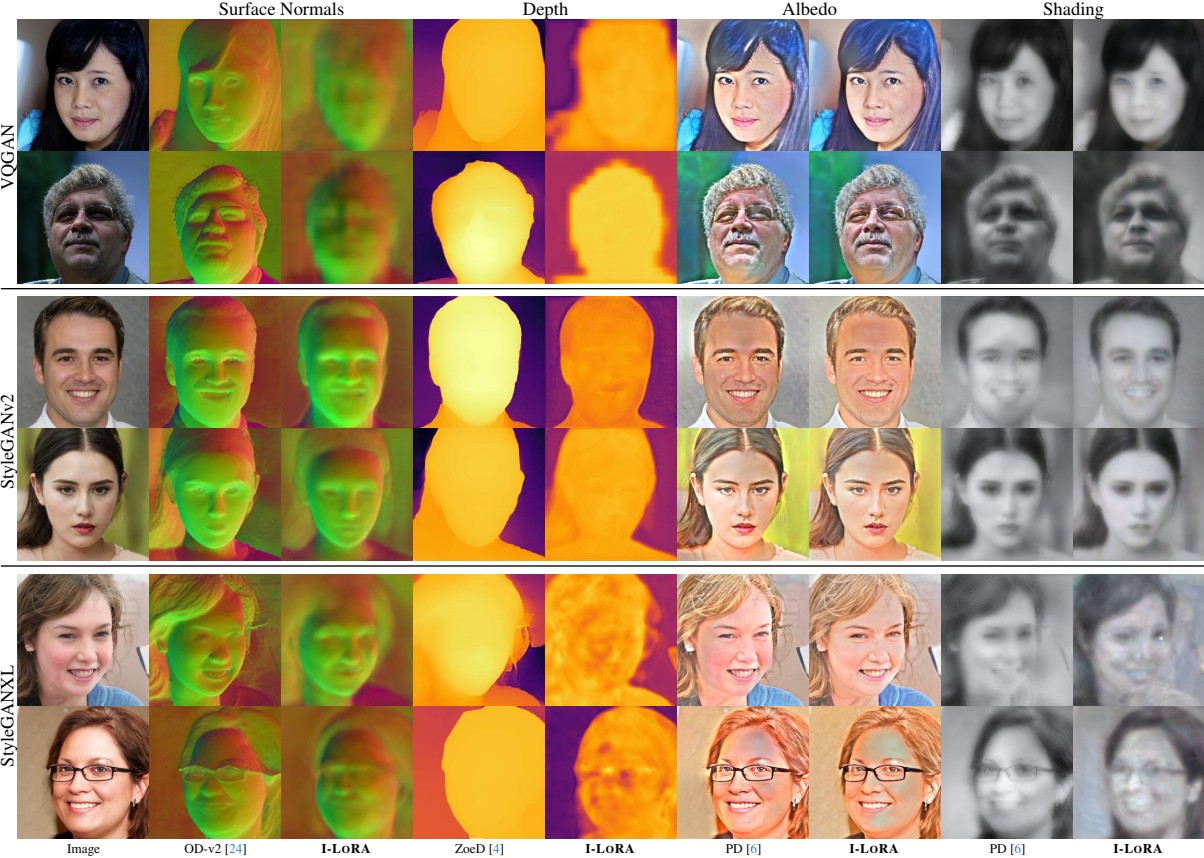

Figure 18. Additional results of scene intrinsics from different generators – VQGAN, StyleGAN-v2, and StyleGAN-XL – trained on FFHQ dataset.

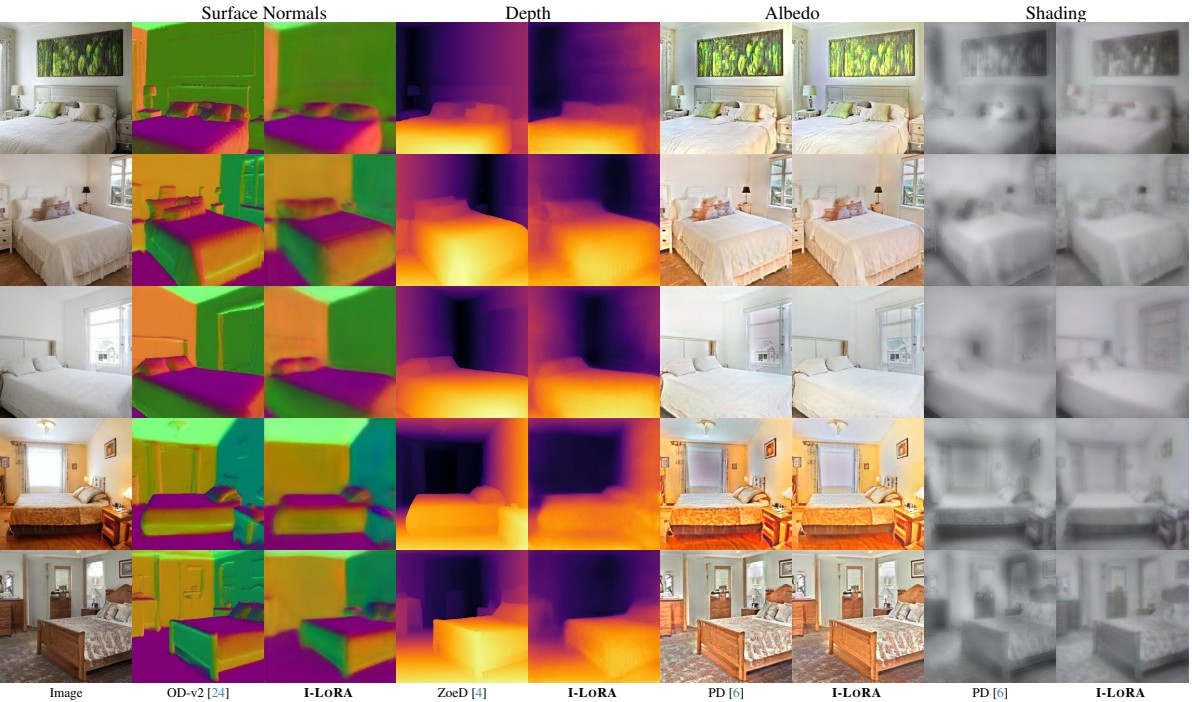

Figure 19. Additional results of scene intrinsics extraction from Stylegan-v2 trained on LSUN bedroom images.

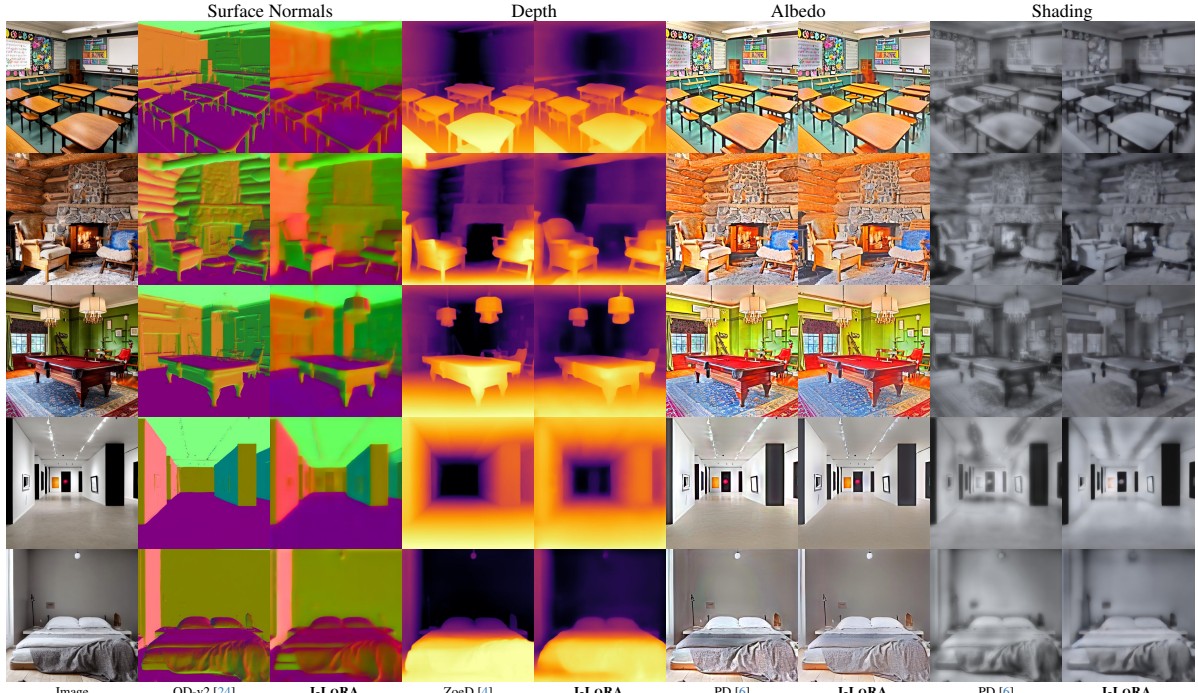

Figure 20. Additional results of scene intrinsics extraction from Stable Diffusion I-LoRA (single-step).

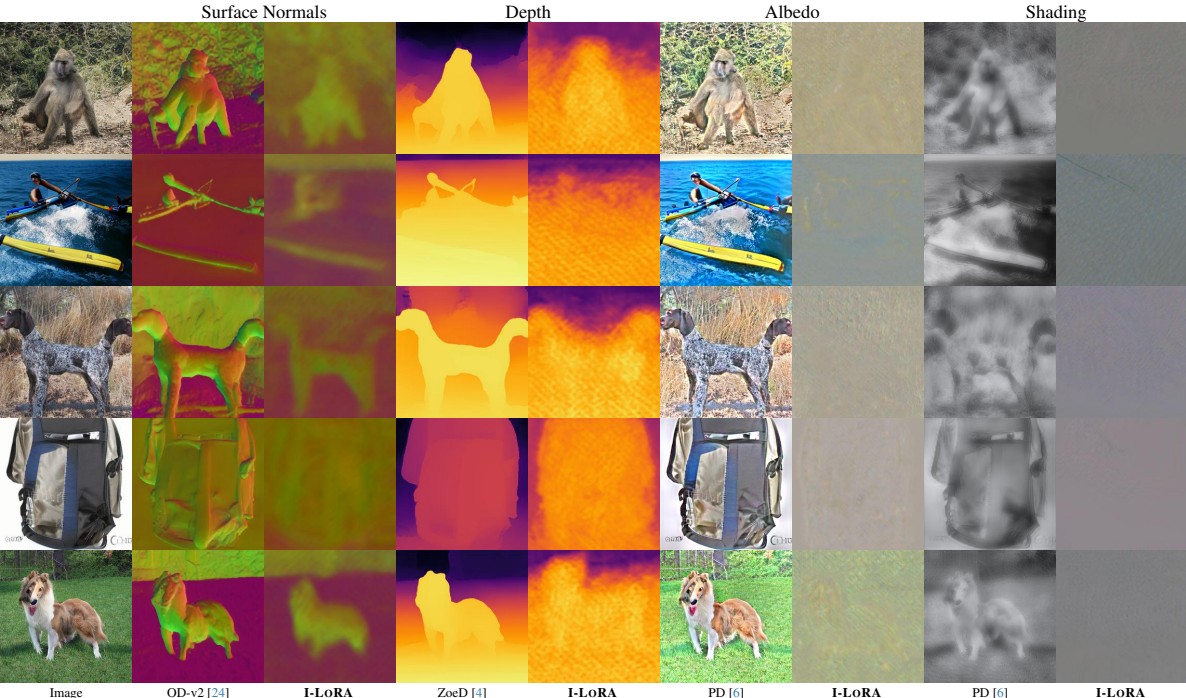

Figure 21. Additional results for StyleGAN-XL trained on ImageNet. StyleGAN-XL's inability to produce image intrinsics may be due to its inability to create high-quality plausible images.

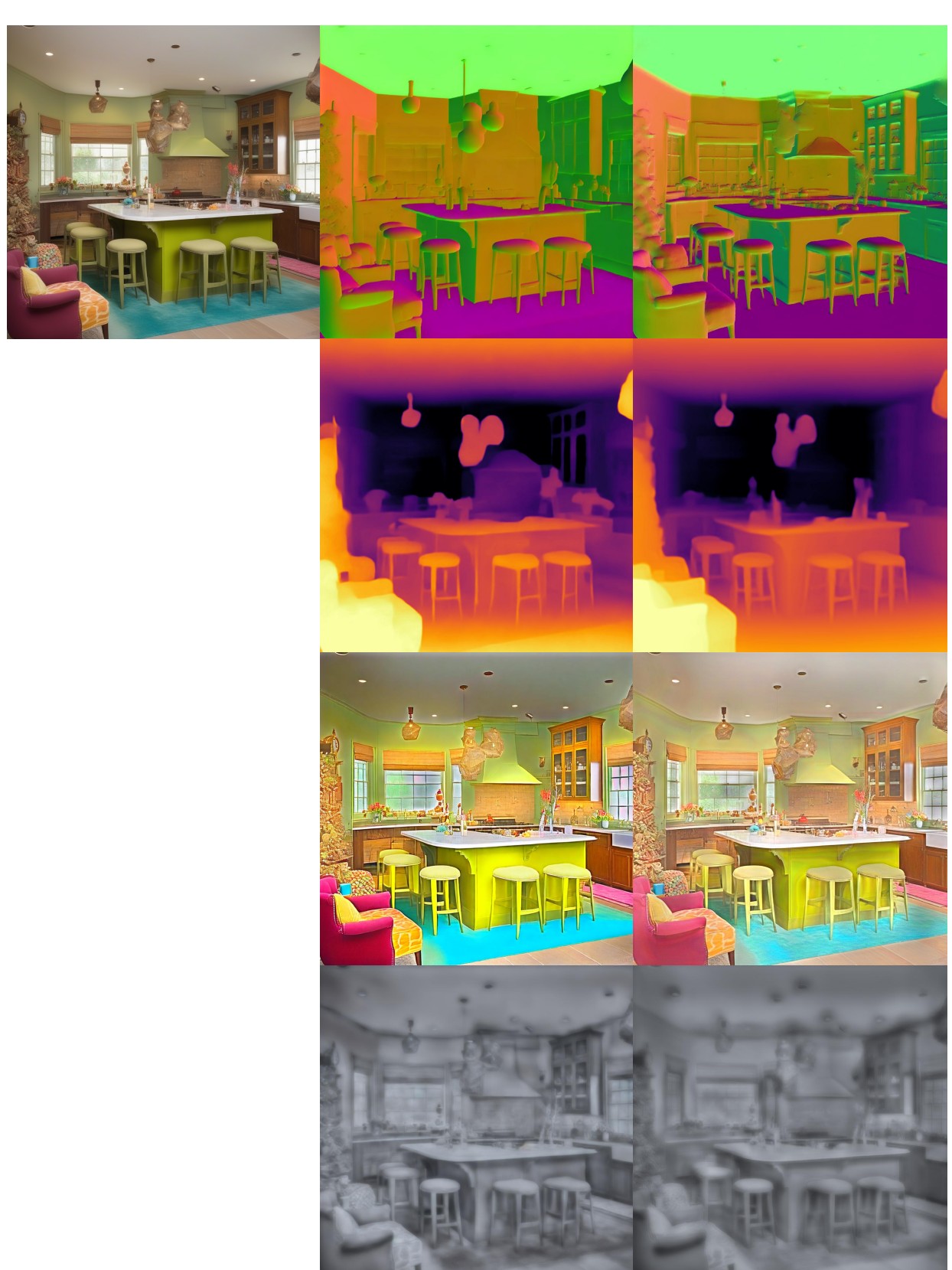

Figure 22. Results of I-LoRA$_{\text{AUG}}$ models applied on unseen $1024^2$ synthetic images. Left: original image; middle: ours; right: pseudo ground truth.

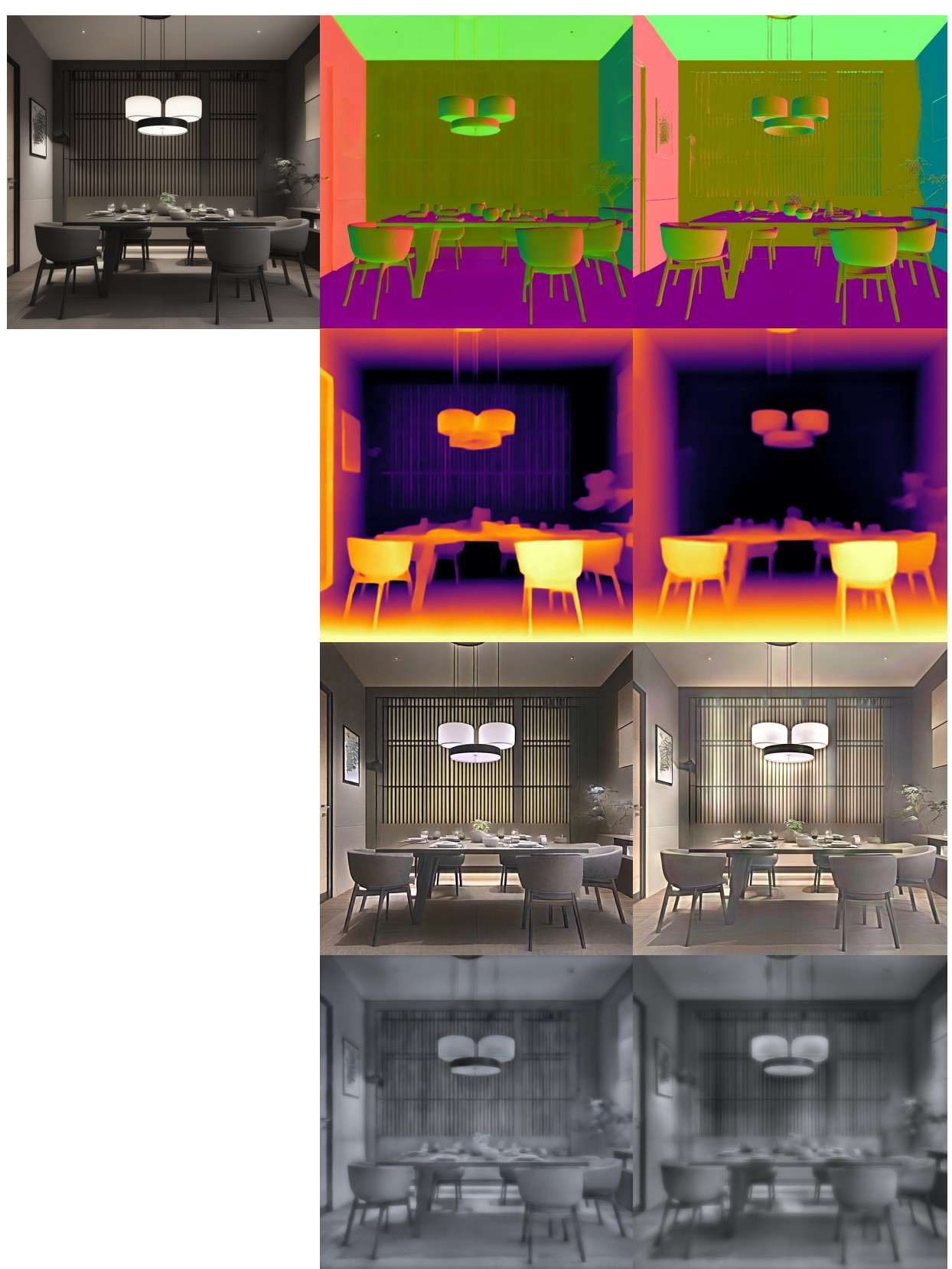

Figure 23. Cont. results of I-LoRA$_{\text{AUG}}$ models applied on unseen $1024^2$ synthetic images. Left: original image; middle: ours; right: pseudo ground truth.

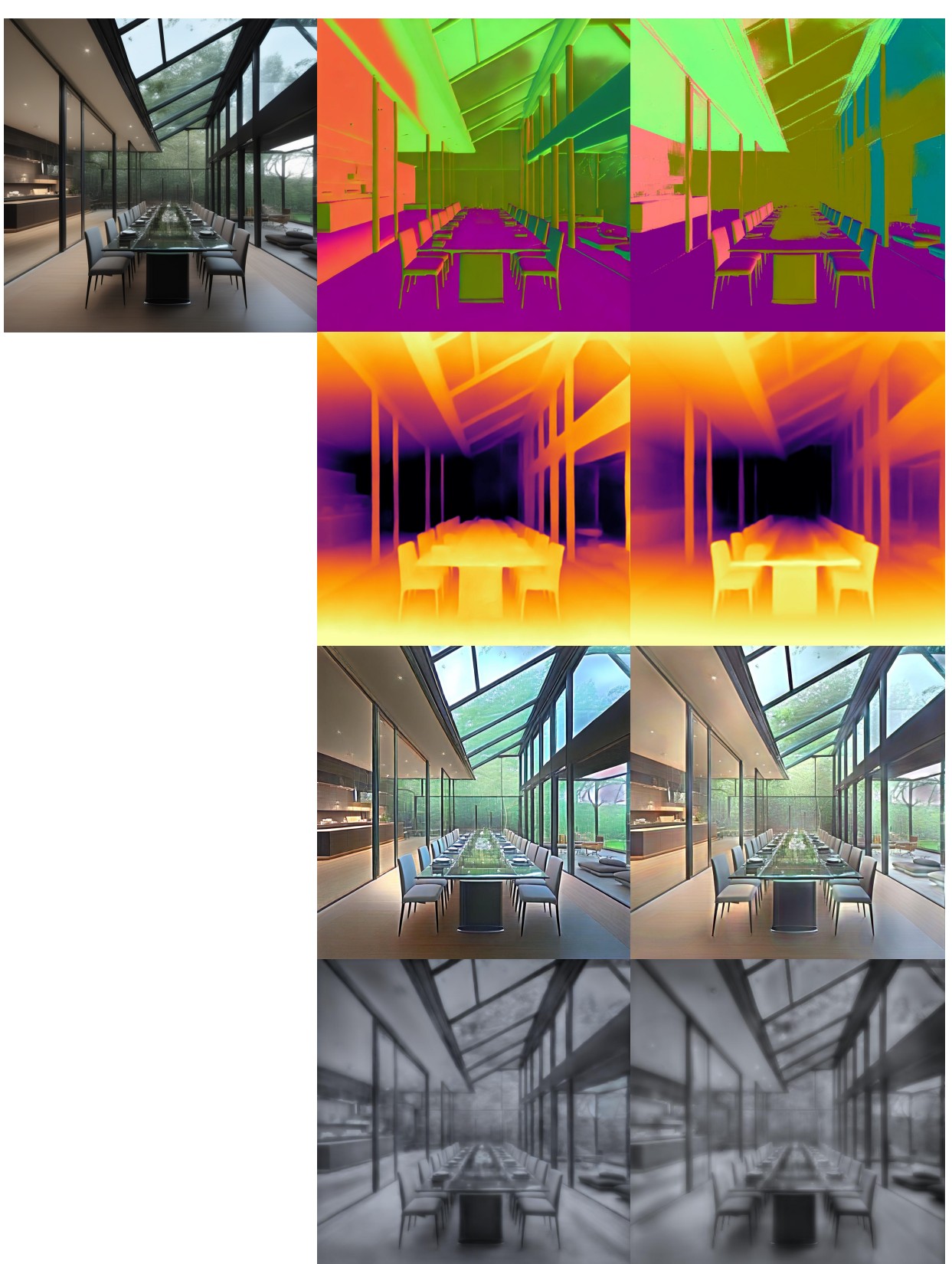

Figure 24. Cont. results of I-LoRA$_{\text{AUG}}$ models applied on unseen $1024^2$ synthetic images. Left: original image; middle: ours; right: pseudo ground truth.

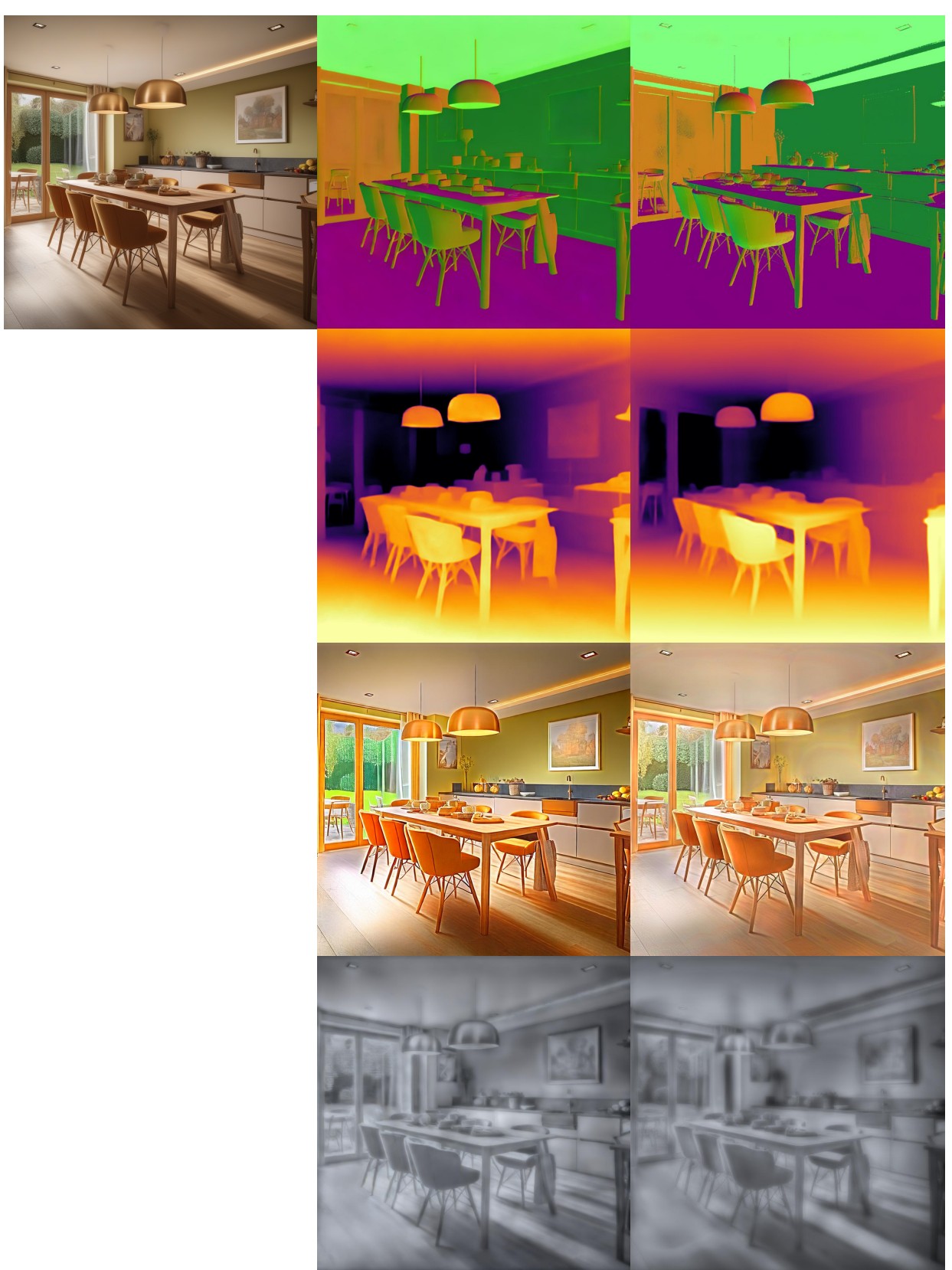

Figure 25. Cont. results of I-LORA_AUG models applied on unseen $1024^2$ synthetic images. Left: original image; middle: ours; right: pseudo ground truth.

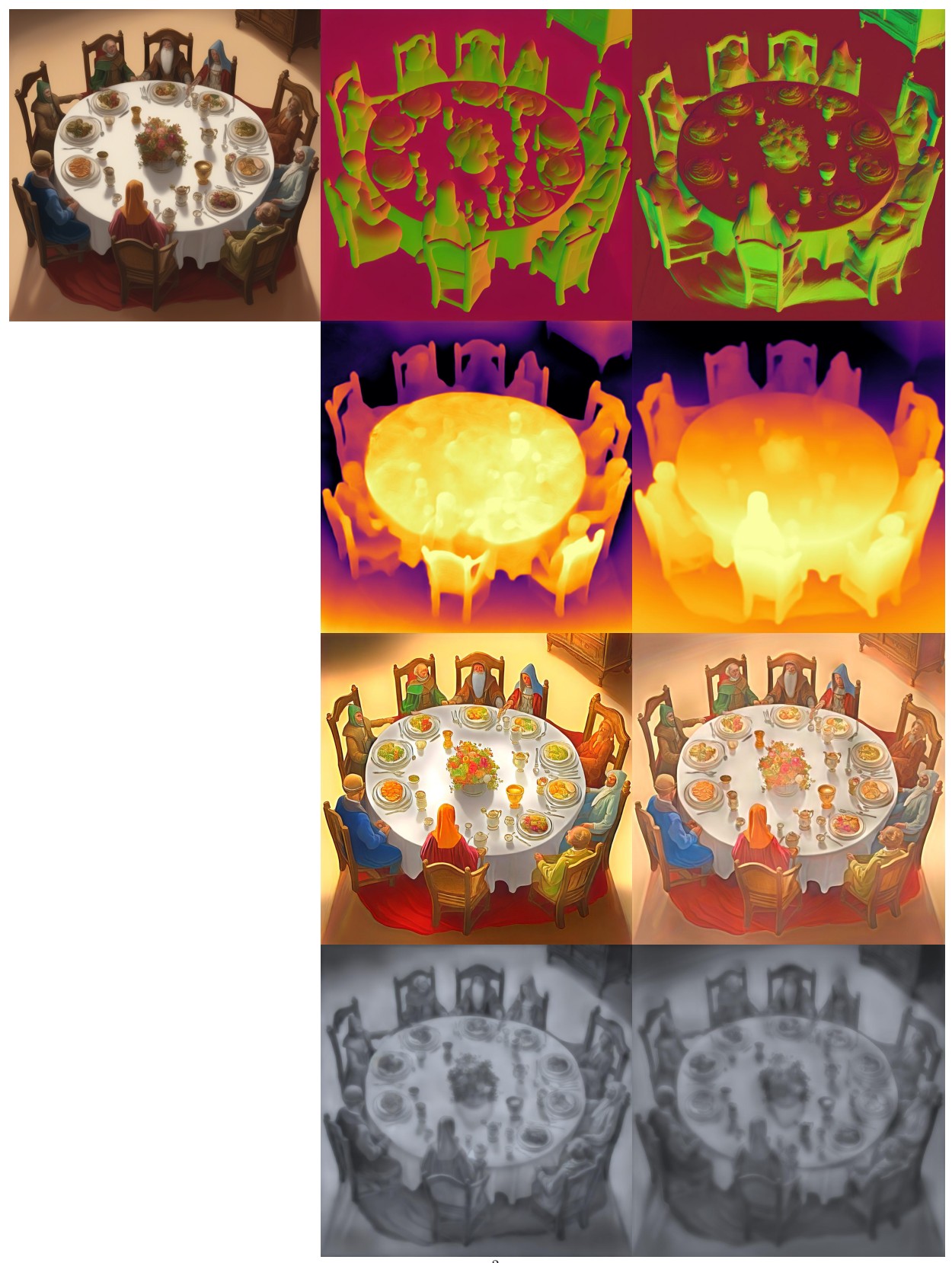

Figure 26. Cont. results of I-LoRA$_{\text{AUG}}$ models applied on unseen $1024^2$ synthetic images. Left: original image; middle: ours; right: pseudo ground truth.

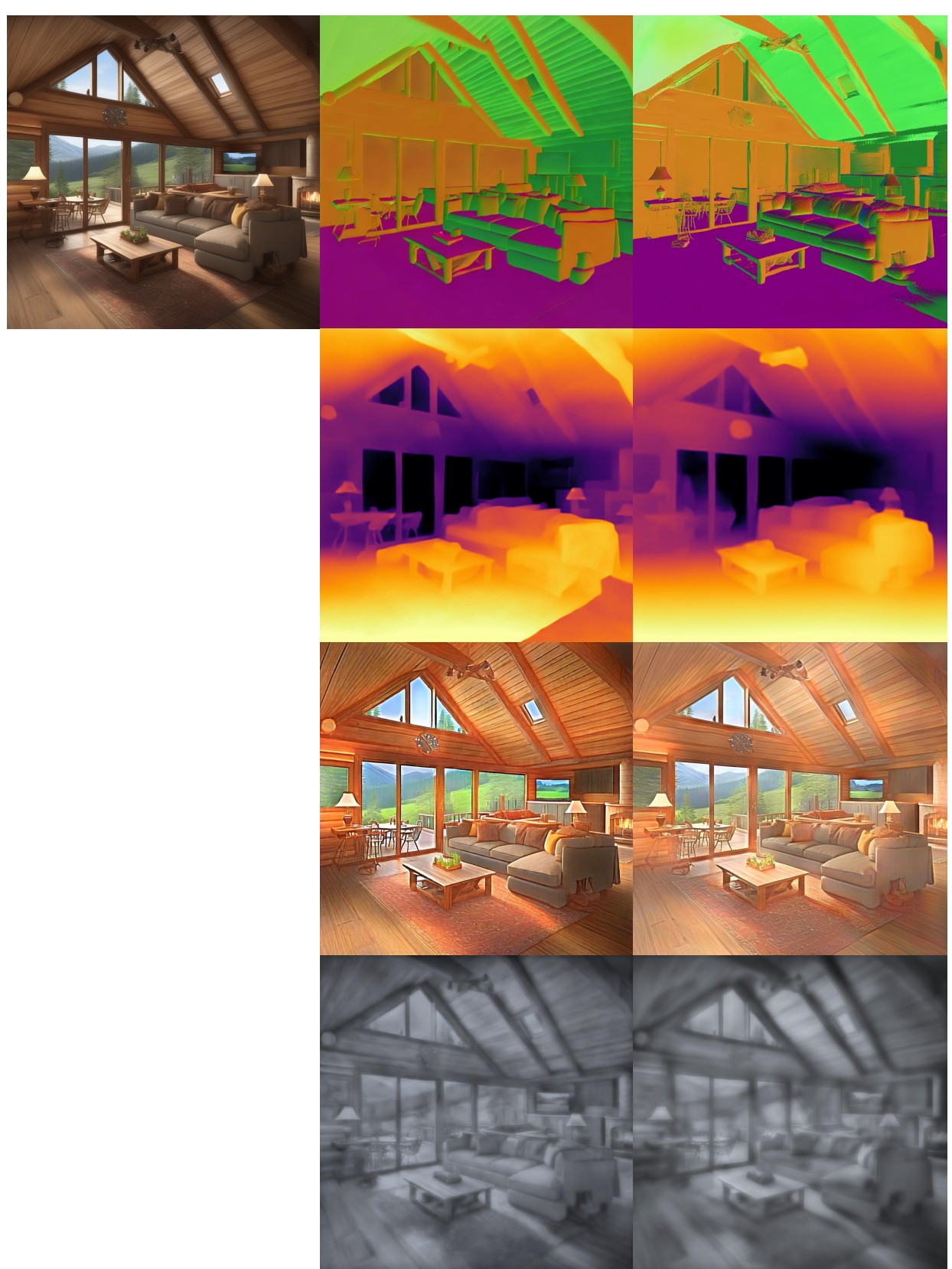

Figure 27. Cont. results of I-LoRA$_{\text{AUG}}$ models applied on unseen $1024^2$ synthetic images. Left: original image; middle: ours; right: pseudo ground truth.

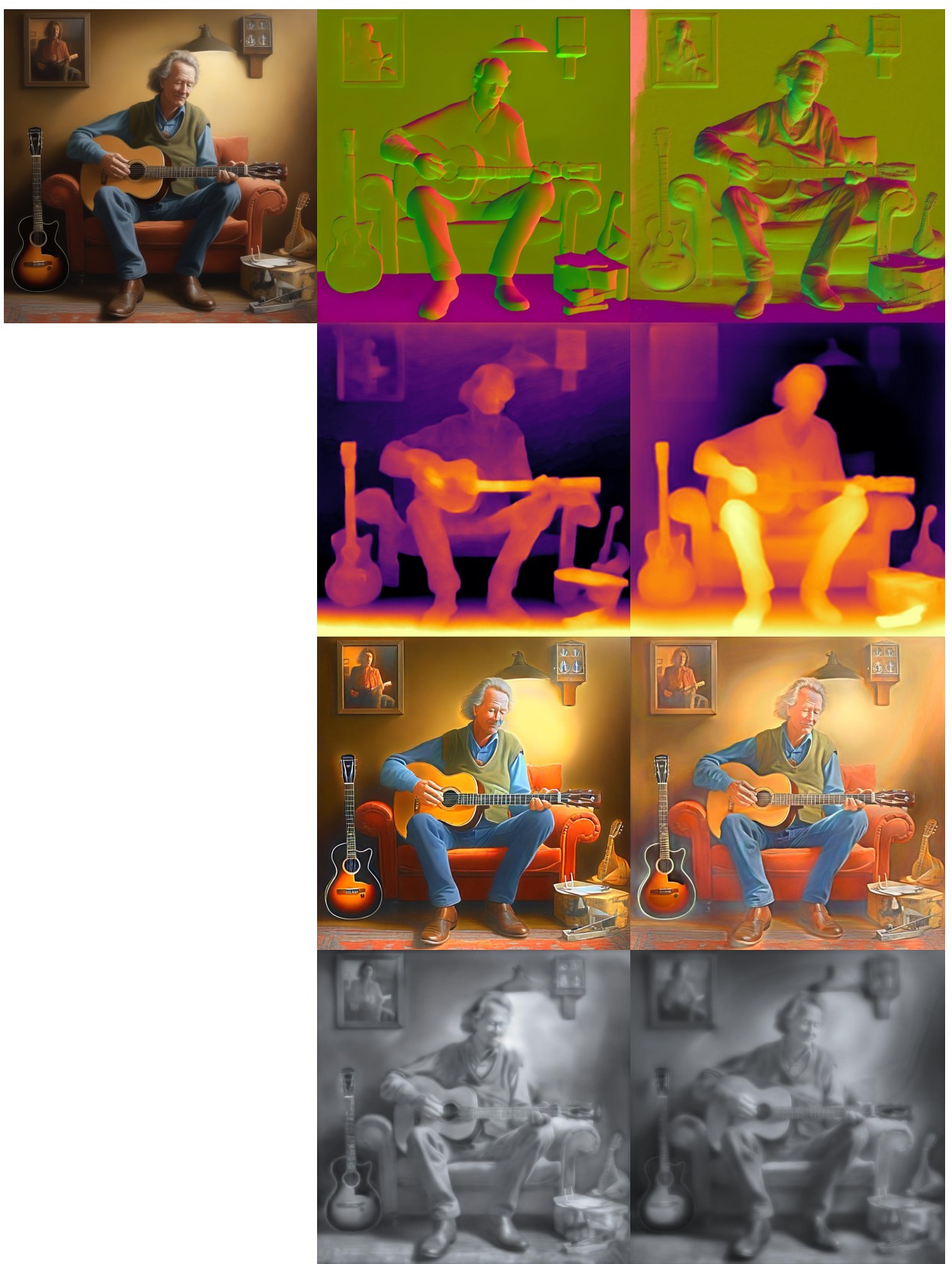

Figure 28. Cont. results of I-LoRA$_{\text{AUG}}$ models applied on unseen $1024^2$ synthetic images. Left: original image; middle: ours; right: pseudo ground truth.

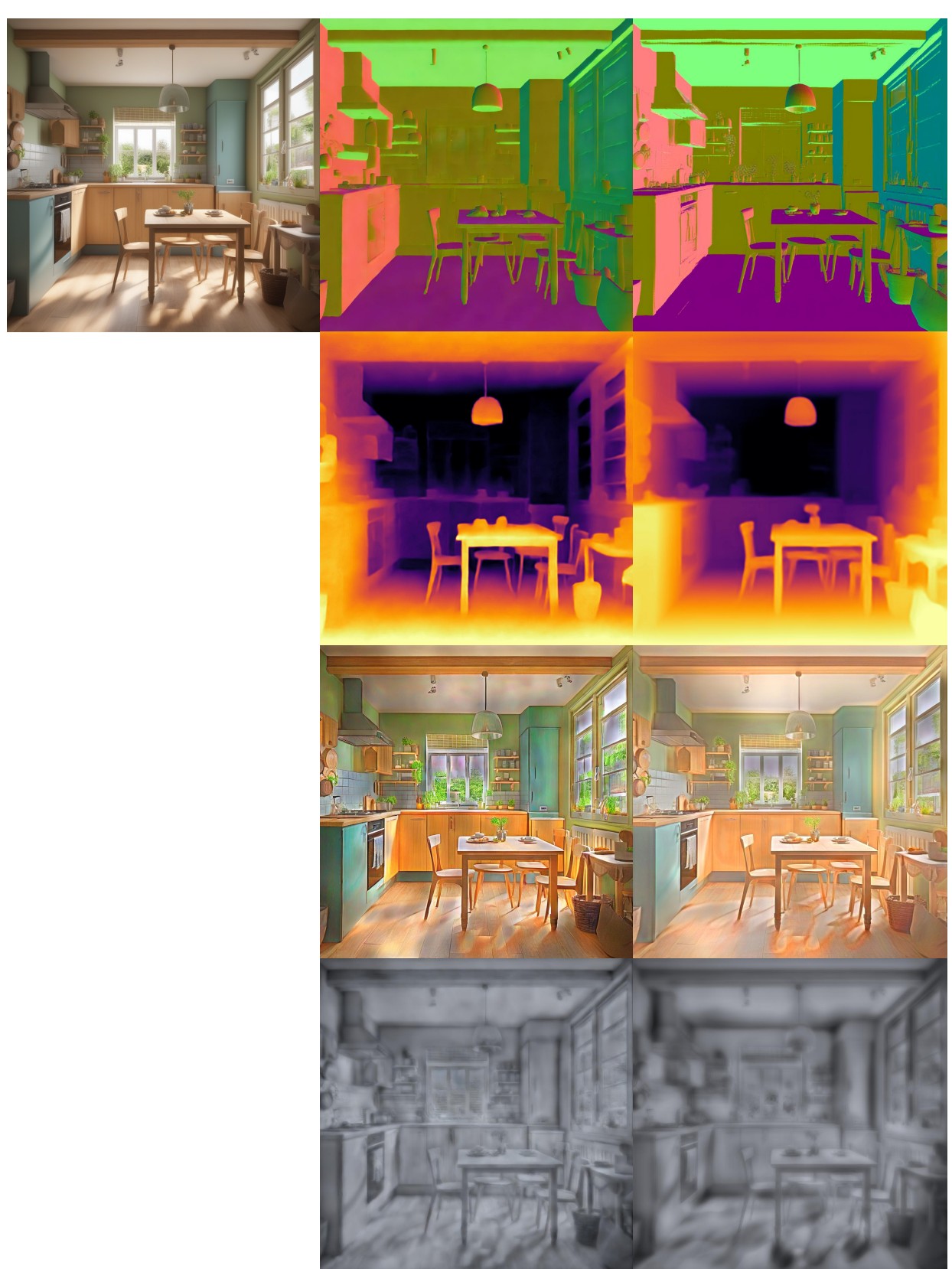

Figure 29. Cont. results of I-LoRA$_{\text{AUG}}$ models applied on unseen $1024^2$ synthetic images. Left: original image; middle: ours; right: pseudo ground truth.

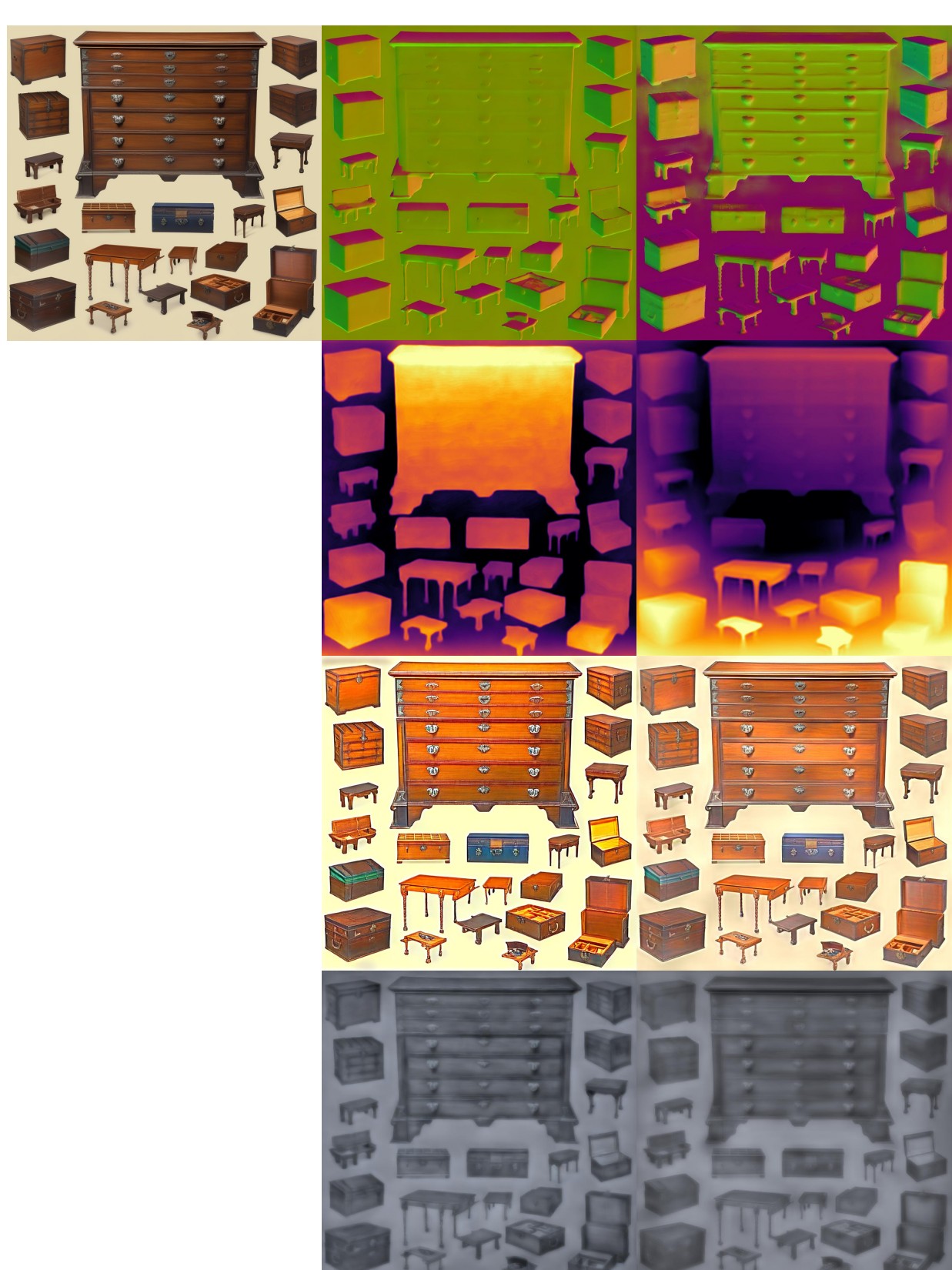

Figure 30. Cont. results of I-LoRA$_{\text{AUG}}$ models applied on unseen $1024^2$ synthetic images. Left: original image; middle: ours; right: pseudo ground truth.

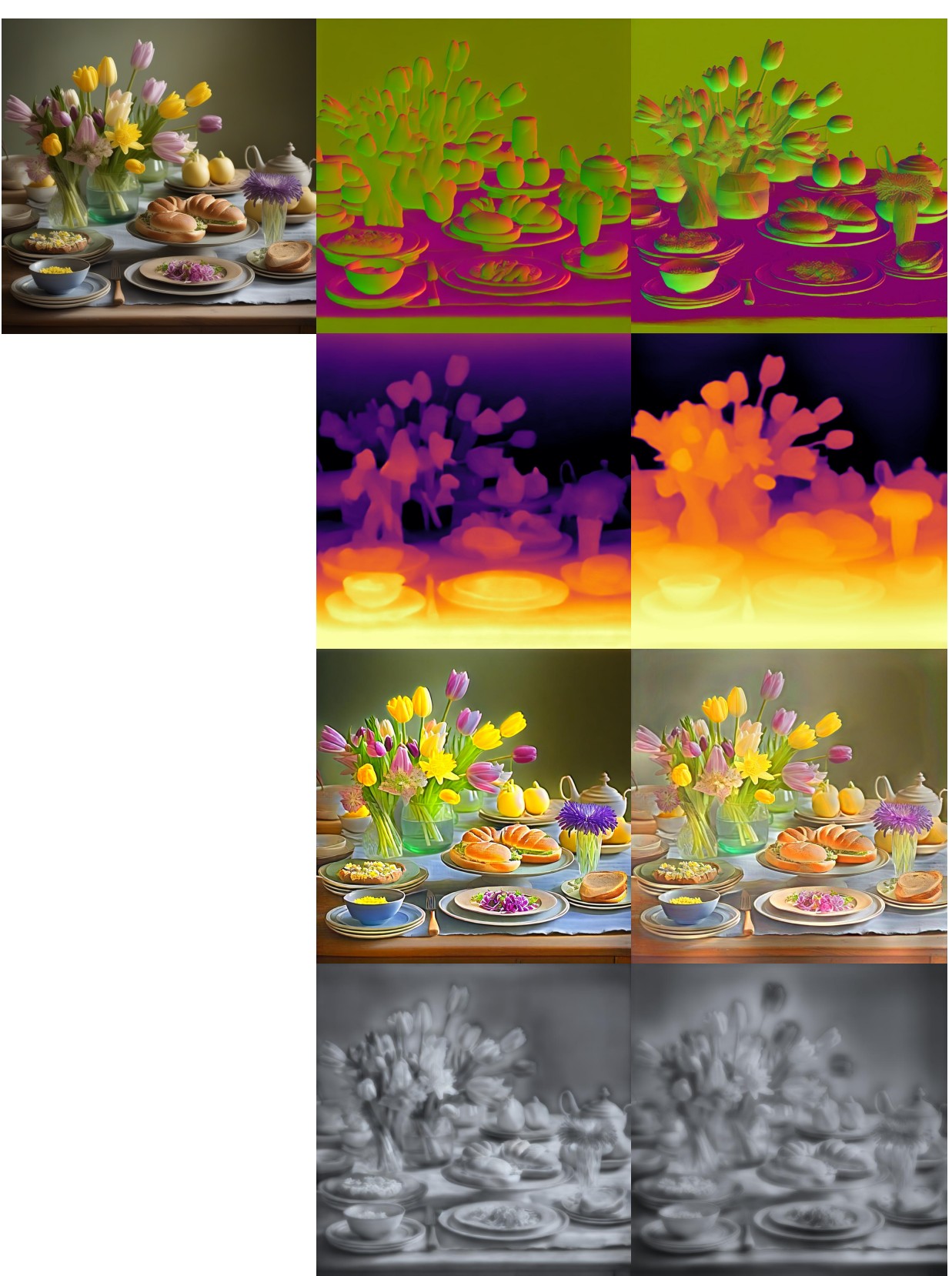

Figure 31. Cont. results of I-LoRA$_{\text{AUG}}$ models applied on unseen $1024^2$ synthetic images. Left: original image; middle: ours; right: pseudo ground truth.