# OpenReview forum: "Intrinsic LoRA: A Generalist Approach for Discovering Knowledge in Generative Models"
_thecvf.com/CVPR/2024/Workshop/SyntaGen — SyntaGen 2024_

### Official Review · Reviewer_sBMs · 2024-03-31
**The paper uses a lightweight LoRA to extract scene intrinsics from various generative model**

**Rating:** 7
**Confidence:** 4

**Review:**

- The paper explores an interesting application of LoRA in extracting scene intrinsics across various generative models.
- It effectively demonstrates the extraction of multiple scene intrinsic attributes such as normals, depth, albedo, and shading, showcasing its wide range of applicability.
- The experiments in the paper cover different types of generative models, stable diffusion versions, multi-step generation, augmentation techniques, and more, which enrich the analysis and strengthen the findings.
- However, a limitation arises because the provided training pipeline requires a ground truth predictor network, which could make it challenging to apply in real-world applications.

---

### Official Review · Reviewer_5NBQ · 2024-03-31
**The idea is interesting and practical, and the results are strong and promising. Thus, I vote for a strong acceptance.**

**Rating:** 9
**Confidence:** 4

**Review:**

## Summary

This paper proposes Intrinsic LoRA (I-LoRA), which can extract visual knowledge from generative models of various types. It achieves this by fine-tuning a pre-trained generative model using LoRA supervised with minimal target datasets. The authors have demonstrated its effectiveness in multiple models like GANs, autoregressive models, and diffusion models. In several metrics, I-LoRA not only matches but also surpasses the performance of its training signals, demonstrating the potential of using generative models as a strong prior for visual knowledge extraction.

## Strength

1. The idea is interesting and practical, as it can be used for many tasks where we cannot collect a large dataset.
2. The experiments are thorough and complete, fully demonstrating I-LoRA's capability and potential.
3. The paper is well-written, and the story is convincing.

## Weakness

1. It is still quite promising to leverage multi-step diffusion inference. The authors may refer to Marigold [1] for any inspirations as they successfully leveraged multi-step diffusion inference for depth estimation.

[1] Repurposing Diffusion-Based Image Generators for Monocular Depth Estimation, CVPR 2024

---

### Official Review · Reviewer_bR7u · 2024-03-31

**Rating:** 7
**Confidence:** 4

**Review:**

### Summary

This work introduces a novel approach named Intrinsic LoRA (I-LORA) which leverages Low-Rank Adaptation (LoRA) to extract scene intrinsics like normals, depth, albedo, and shading from a variety of generative models. I-LORA is highlighted for its efficiency, requiring only minimal extra parameters and training data. It has been tested across different generative models and datasets, demonstrating broad applicability and the ability to extract scene intrinsics with competitive quality compared to leading supervised techniques.

### Strengths
+ The finding that well-trained generative networks, **even** autoregressive models, have some implicit knowledge about scene intrinsics is very interesting. I believe that prior works have not yet showcased this phenomenon as they only focus on a specific type of generative models.

+ I-LORA's design minimizes the additional computational overhead and parameter count, making it an efficient method for enhancing generative models with the ability to extract scene intrinsics.

### Weaknesses
+ In some case, this work assumpt to have a few labels predicted by a pretrained models, which is quite unrealistic. This is only minor criticism as we can manually creates the labels. Furthermore, the author claims that I-LoRA is data-efficient but do not report how much labels are required.

+ The effectiveness of I-LORA appears to be model-dependent as the layers, to which LoRA is applied, are different for each type of models (L147-153). It is a bit unclear to me why the author chooses these specific layers and how robust I-LoRA is if different layers are chosen.

### Questions
+ Why introduce Augmented I-LoRA if it does not bring any improvements ?
+ How I-LoRA produces the scene intrinsics is unclear to me. Is it that the added LoRA parameters predict the scene intrinsics? Or are these intrinsics predicted at the final layer alongside with the final output ?

---

### Decision · Program_Chairs · 2024-04-06

**Decision:**

Accept

**Comment:**

All three reviewers strongly support paper acceptance due to the paper's interesting algorithm and discovery, its efficient design, wide range of applicability, comprehensive experiments, clear writing, and convincing story. Some weaknesses are mentioned but do not affect the acceptance decision.

The ACs agree to accept the paper. The authors should consider the reviewers' suggestions to further improve the work.